

# Measurement report: High Arctic aerosol hygroscopicity at sub- and supersaturated conditions during spring and summer

Andreas Massling[1], Robert Lange[1,2], Jakob Boyd Pernov[1,3], Ulrich Gosewinkel[1], Lise-Lotte Sørensen[1], and Henrik Skov[1]

[1]Department of Environmental Science, iClimate, Aarhus University, 4000 Roskilde, Denmark
[2]ROCKWOOL Group, 2640 Hedehusene, Denmark
[3]Extreme Environments Research Laboratory, École Polytechnique Fédérale de Lausanne, 1951 Sion, Switzerland

*Correspondence to*: Andreas Massling (anma@envs.au.dk)

**Abstract.** Aerosol hygroscopic growth and cloud droplet formation influence the radiation transfer budget of the atmosphere and thereby the climate. In the Arctic, these aerosol properties may have a more pronounced effect on the climate compared to the mid-latitudes. Hygroscopic growth and cloud condensation nuclei (CCN) concentrations of High Arctic aerosols were measured during two field studies in the spring and summer of 2016. The study site was the Villum 
Research Station (Villum) at Station Nord in the northeastern region of Greenland. Aerosol hygroscopic growth was measured with a hygroscopic tandem differential mobility analyzer (HTDMA) over a total of 23 days, and CCN concentrations were measured over a period of 95 days. Continuous particle number size distributions were recorded, facilitating calculations of aerosol CCN activation diameters and aerosol kappa ($\kappa$)-values. In spring, average CCN concentrations, at supersaturations (SS) of 0.1 to 0.3 %, ranged from 53.7 to 85.3 cm$^{-3}$, with critical activation diameters 
ranging from 130.2 to 80.2 nm, and $\kappa_{CCN}$ ranging from 0.28-0.35. In summer, average CCN concentrations were 20.8 to 47.6 cm$^{-3}$, while critical activation diameters and $\kappa_{CCN}$ were from 137.1 to 76.7 nm and 0.23-0.35, respectively. Mean particle hygroscopic growth factors ranged from 1.60 to 1.75 at 90% relative humidity in spring, while values between 1.47 and 1.67 were observed in summer depending on initial dry size. Although the summer aerosol number size distributions were characterized by frequent new particle formation events, the CCN population at cloud-relevant 
supersaturations was determined by accumulation mode aerosols. This emphasizes the importance of accumulation mode aerosol sources to provide available CCN during summer. The influence of particle hygroscopic growth on the radiative transfer through aerosol-radiation interactions could be of major importance. The results of this study are directly applicable in the modeling of direct and indirect climate effects of Arctic aerosols. Targeted chemical and morphological analysis, based on filter samples or on-line techniques, could further clarify the role of primary organic marine influence 
on Arctic aerosol CCN concentrations and therewith climate effects.





## 1 Introduction:

In the Arctic region, the average temperature is rising about three times as fast as the global average (Lenssen et al., 2019). Aerosols influence the radiation balance of the atmosphere and the surface and thus play a key role in the climatic changes observed in the Arctic (Bellouin et al., 2011; Yang et al., 2014; Chylek et al., 2016). During sunlit seasons, the climatic effects of Arctic aerosols are diverse. Through aerosol-radiation interactions, aerosols scatter incoming shortwave radiation and absorb outgoing longwave radiation (Twomey, 1974). Arctic aerosols also impact clouds by altering cloud lifetime, precipitation rate, and cloud albedo (Zhao and Garrett, 2015). These aerosol-cloud interactions also influence the radiation balance. It has been shown that the Arctic atmosphere is limited in cloud condensation nuclei (CCN) (Mauritsen et al., 2011). This means that a relatively small increase in the concentration of aerosol particles that can function as CCN, can have a comparably large effect on clouds and thus the radiation balance. The radiative forcing (RF) of aerosol-radiation interactions in the Arctic is not well quantified but appears to be a net cooling of the surface (Quinn et al., 2008; Breider et al., 2017).

Aerosol light scattering is highly dependent on particle size and refractive index (Moffet and Prather, 2009; Raut et al., 2009). Particles interact with atmospheric water vapor and potentially grow due to hygroscopicity (or ability to uptake water) (Pilat and Charlson, 1966; Carrico et al., 2000; Fierz-Schmidhauser et al., 2010). This water uptake at subsaturated conditions changes their light scattering behavior (Liu et al., 2013; Titos et al., 2016). One parameter to deduct from hygroscopicity measurements of aerosols is the kappa-value ($\kappa$-value), which provides a theoretical framework for deriving the hygroscopicity for bulk aerosols, assuming an internal mixture (Petters and Kreidenweis, 2007). Knowledge about the hygroscopic growth of Arctic aerosols is crucial for determining the effects of their aerosol-radiation and aerosol-cloud interactions.

Ambient Arctic aerosol size, chemical composition, and number concentrations undergo a strong seasonal cycle (Tunved et al., 2013; Nguyen et al., 2016; Freud et al., 2017). Very few anthropogenic aerosol sources exist within the Arctic. During summer, the Arctic lower troposphere is a largely isolated system (Iversen and Joranger, 1985; Stohl, 2006). Anthropogenic influence from long-range transport is mostly absent, although intrusion from high latitude regions, introducing anthropogenic aerosols, does occur. In summer, the ambient aerosol is mainly of biogenic origin, consisting mostly of biogenic sulfate and organic compounds (Willis et al., 2017). The aerosol is characterized by low particle number concentrations (50-200 cm$^{-3}$), interrupted by relatively frequent new particle formation (NPF) events which strongly increase the particle number concentration on timescales of hours to days (Nguyen et al., 2016; Freud et al., 2017; Collins et al., 2017; Dall'Osto et al., 2017). During autumn months, the occurrence of NPF decreases, and particle number concentrations reach a minimum. During winter, the polar front moves southward, extending the Arctic lower tropospheric system to include mid-latitude regions with anthropogenic emissions (Shaw et al., 2010). During the winter, inefficient wet deposition coupled with a stably stratified atmosphere allows for pollution to build up forming an increasingly prominent accumulation mode fraction of aerosols. This accumulation mode reaches its maximum number concentration in late spring (March-April) (Matsui et al., 2011; Freud et al., 2017; Lange et al., 2018), a phenomenon known as Arctic haze (Barrie et al., 1981; Shaw, 1995; Quinn et al., 2007). During the Arctic winter and early spring, when sunlight is absent or scarce, it has been demonstrated that especially anthropogenic aerosols have a warming effect due to resulting increases in cloud long-wave emissivity (Garrett et al., 2004; Garrett and Zhao, 2006). During this period, sulfate is the dominating particle component (Heidam et al., 1999; Heidam et al., 2004; Nielsen et al. 2019). Recent findings indicate that sulfate, sea spray aerosol and organic carbon might be internally mixed also in the submicron size range (Kirpes et al., 2018). When temperatures increase around May, the polar dome recedes and wet scavenging becomes an important aerosol sink, consequently, removing the Arctic haze (Engvall et al., 2008; Browse et al., 2012; Croft et al., 2016). This leads to the reestablishment of clean summer conditions.

Concentrations and temporal development of CCN in the Arctic are characterized in several previous studies. Silvergren et al. (2014) evaluated a full CCN seasonal cycle (2007-2008) on Zeppelin Mountain, Svalbard, from re-aerosolized filter sampled aerosol. For supersaturations (SS) between 0.2–0.5 %, they found low CCN concentrations of < 40 cm$^{-3}$ for months September–October, ~ 40–80 cm$^{-3}$ for November–February, 80–160 cm$^{-3}$ for March–June and 40–110 cm$^{-3}$ for July–August, with the highest variation between supersaturations in the latter period. In contrast, Zabori et al. (2015) directly determined $\kappa$-values from size-resolved CCN measurements on Svalbard to be between 0.4 and 0.3 at 0.4 % SS



on separate occasions in June and August 2008, respectively. A review of long-term studies from around the world, including Utqiaġvik/Barrow, Alaska from 2007 to 2008, was published by Schmale et al. (2018). CCN concentrations at 0.2 % SS were between 20 and 200 cm$^{-3}$ and CCN particle critical activation diameters (Dp$_{crit}$) showed a bimodal distribution, with a smaller mode at 70–100 nm and a larger one at 100–140 nm. The location showed the highest CCN number concentrations and lowest Dp$_{crit}$ from February to June. In a study conducted in Northwestern Canada in May

2014, Herenz et al. (2018) found CCN concentrations between 10 and 200 cm$^{-3}$ at 0.1–0.78 % SS and found a κ-value of 0.19 at 0.1 % SS and κ-values of 0.21–0.28 at 0.2–0.7 % SS. They detected both well-aged Arctic haze aerosols, as well as more recently formed aerosols, characteristic of the Arctic summer period.

From late spring to summer, the Arctic aerosol organic fraction changes from having a strong anthropogenic influence to being mainly biogenic (Nielsen et al., 2019; Willis et al., 2017). Likewise, the total organic fraction increases during

summer as well (Chang et al., 2011; Breider et al., 2014). The origin and influence of these organic components on aerosol CCN ability (Leck et al., 2002) and hygroscopic behavior are currently not well understood and are still the subject of intense research (Myriokefalitakis et al., 2010; Fu et al., 2013; Willis et al., 2016). An organic marine source has been proposed to be an important source for primary aerosols, NPF, and secondary growth (Leck and Bigg, 2005; Burkart et al., 2017; Willis et al., 2017), which contributes significantly to the summertime aerosol mass load. Methanesulfonic acid

(MSA) (Nielsen et al. 2019; Dall'Osto et al., 2018b) and organic acids (Mungall et al., 2017), high molecular weight compounds like proteins (Fu et al., 2015), and polymeric gels (Orellana et al., 2011) likewise play a role. Further, inorganic non-sea salt sulfate from the oxidation of dimethyl sulfide (DMS) contributes importantly to aerosol mass during summer and autumn (Rempillo et al., 2011; Leaitch et al., 2013). Surface active organics potentially increase aerosol CCN activity by lowering droplet surface tension (Lohmann and Leck, 2005). However, studies on non-Arctic

marine aerosols at Mace Head have suggested a complex behavior, where biogenic marine particles with low hygroscopic growth featured high CCN activity (Ovadnevaite et al., 2011). This behavior can potentially be explained by liquid phase separation, where less-hygroscopic but surface-active components are located in an outer shell limiting hygroscopic growth at subsaturated conditions. During CCN activation, the phase separation breaks down and the surface-active compounds enhance CCN activity (Ovadnevaite et al., 2017). This process could also be relevant for Arctic aerosols.

However, Lange et al. (2018) found indications that surface activity is not necessary to explain κ-values of Arctic accumulation mode aerosols.

Longer field studies reporting hygroscopic growth measured by HTDMA instruments in the Arctic are scarce. Zhou et al. (2001) conducted a shipboard campaign in the Arctic Ocean up to 87° 30' N during the summer of 1996. They reported hygroscopic growth factors (HGF) of 1.4–1.9 at 90 % relative humidity (RH) to be most prominent, however, cases of

HGF >1.9 and HGF <1.4 were also observed. The higher HGFs probably result from sea-salt aerosol, while the lower HGF could originate from an organic sea surface source. Using the same measurement series, Silvergren et al. (2014) determined HGF-derived κ$_{HTDMA}$-values to be 0.35–0.5 in March-August and 0.6–0.9 in September–January. During a ship cruise in July–August 2013, Allan et al. (2015) observed that a possible contribution from iodine oxides lowered the HGF of 50 nm particles at 90% RH to 1.34, whereas HGF otherwise was consistently >1.5.

The present study evaluates measurements from two field studies in 2016, where CCN concentrations and HGFs were measured at the High Arctic site, Villum Research Station (Villum). We present the measurement results and evaluate which aerosol sources could be responsible for the observed findings at subsaturated and supersaturated conditions. Our results can directly be used in models of Arctic aerosol–cloud–climate interactions, while simultaneously adding to the understanding of which aerosol sources are important for Arctic CCN concentrations.


## 2  Methods

### 2.1  Sampling site

All measurements presented in this study were recorded at the Villum Research Station (81° 36' N, 16° 40' W) in northeast Greenland. Its location is shown in Fig. 1. Villum is located at the Danish military station of Station Nord on the Princess

Ingeborg Peninsula, about 750 m from the coast of Denmark Fjord, close to the Wandel Sea. This coastal site is within



the polar dome year-round and is in close proximity to the Arctic pack ice during both spring and summer (Fig. 1). Currently, the entrance from the Wandel Sea into the Denmark Fjord is covered by sea ice year-round, although in recent years, the waters near the station are unfrozen from August to September. The sampling took place 2 km southwest of the station's main facilities in the "Air Observatory" which provides controlled laboratory conditions for instrument

operation. The location is upwind of the main station premises >95 % of the time, pollution influx from the station itself is therefore minimal. Polar sunrise occurs on February 25th, polar day prevails from April 5th to September 3rd and polar sunset occurs on October 16th. From October 16th to February 25th, the station experiences polar night. Further description of the station is found elsewhere (Nguyen et al., 2016; Lange et al., 2018).



**Figure 1. Location of Villum Research Station at Station Nord.** Villum is shown with a red star. Average sea-ice concentration is included for the spring campaign (left) and summer campaign (right). Sea-ice concentrations were taken from the National Sea & Ice Data Center (https://nsidc.org/data/seaice_index/archives/).

The field studies described in this work were conducted in 2016 over two campaigns, one in spring and one in summer.
The exact periods when the field studies were carried out are listed in Table 1 below. Basic meteorological parameters (temperature, wind speed and direction, atmospheric pressure, RH and solar radiation) are monitored at Villum.

**Table 1. Measurement periods during spring and summer field study.**

| Instruments | Spring 2016 | Summer 2016 |
|---|---|---|
| CCN | 20 Apr – 6 June | 15 Aug – 2 Oct |
| HTDMA | 22 Apr – 2 May | 15 Aug – 28 Aug |
| SMPS | Full coverage | Full coverage |

Although the duration of the spring field study arguably extends into the meteorological summer, and the summer field study extends into the meteorological autumn, the naming is kept for convenience.



### 2.2 Aerosol number size distributions

A scanning mobility particle sizer (SMPS) has been operated continuously at Villum since 2010. Studies of aerosol physical properties at Villum, including the measurement setup, have been published elsewhere (Nguyen et al., 2016; Freud et al., 2017; Lange et al., 2018, 2019). In brief, the SMPS system measures size distributions of particles with electrical mobility diameters in the range from 9 to 915 nm, with 5 min time resolution. It consists of a custom-built electrostatic classifier with a Vienna type medium DMA column, operating with a 5:1 sheath-to-sample flow ratio

(Wiedensohler et al., 2012), either with a TSI 3010 condensation particle counter (CPC) or TSI 3772 CPC for particle detection. Ambient aerosol is sampled through a specially designed heated inlet for total suspended particles, with laminar flow to minimize losses. There is no additional drying of the sample air prior to entering the analytical instruments, as the transition from the cold outside temperatures to the >20 °C inside of the "Air Observatory" ensures sufficient particle drying. The aerosol sample flow RH is below 35% more than 99% of the time. Raw SMPS measurements are inverted

offline by an algorithm as described in Pfeifer et al. (2014).

All measurements recorded by the SMPS were quality-controlled by inspection of instrument flows, temperatures, and RHs as well as by visual inspection of size distribution plots. Measurements influenced by local pollution from e.g., nearby vehicles, or when the instrument was malfunctioning, were removed from further consideration.

### 2.3 Cloud condensation nuclei properties

Concentrations of CCN were measured by a Droplet Measurement Technologies CCN-100 CCN counter, cycling ten supersaturations of 0.1-1.0 % SS. The highest reachable supersaturation (typically ~1.8 % SS) was also included in the measurement protocol. CCN concentrations at the highest SS ($CCN_{max}$) were used as a reference against the number concentration of particles larger than 25 nm ($N_{25}$) inferred from SMPS measurements. For all settings of SS, the

temperature gradient in the CCN column was allowed to stabilize for 5 min, before acquiring data for 5 min. An exception to this procedure was the return from ~1.8 % SS to 0.1 % SS, where a 15 min stabilization time was allowed. The resulting cycling time for all 11 supersaturations was 120 min. Measurements at 0.25, 0.35 and 0.40 % SS showed to be largely redundant with measurements at other SSs and are therefore not presented further here. The SS setting of the instrument was calibrated with monodisperse ammonium sulfate (AS) aerosol, produced by an atomizer and a scanning electrical

mobility spectrometer (SEMS), both from Brechtel. By varying AS aerosol particle diameter (Dp), $Dp_{crit}$ was determined for 0.1–0.47 % SS at the sampling site four times during the field studies. Subsequently, calibrations were made at 0.1–1.0 % SS in a more rigorous procedure in our laboratory at the Department of Environmental Science at Aarhus University, Risø Campus, Roskilde, Denmark. The combined experimentally obtained values for $Dp_{crit}$ were compared with the corresponding calculated values obtained using the Köhler equation (Kohler, 1936) and AS bulk solution

properties derived from the E-AIM model (Clegg et al., 1992; Wexler and Clegg, 2002). Hereby, a linear relationship between the SS set on the CCN counter ($SS_{set}$) and the real SS value ($SS_{real}$) based on theory was established in the full range of 0.1–1.0 % SS (Least squares fit: $SS_{real} = 0.9019*SS_{set} + 0.0629$, $SD_{SS} = 0.031$ %, $r^2 = 0.995$), applying a method similar to that used by Kristensen et al. (2012), and Nakao et al. (2014). The linear calibration curve is shown in supplementary material 1 (Fig. S1). For convenience, the reported SS values in this paper are the $SS_{set}$ values. Quality

control of CCN data was based on the achievement and stability of the CCN column temperature gradient, and the stability of the instrument temperatures and flows. Additionally, CCN measurements recorded simultaneously with SMPS measurements that were discarded during quality control were removed from further consideration.

Aerosol $Dp_{crit}$ was determined by sequential integration of simultaneously measured particle number size distributions and CCN concentrations at a given SS, decreasing $Dp_{crit}$ until the following relationship was satisfied.

$$\int_{Dp_{crit}}^{Dp_{max}} n_N(Dp)dDp = CCN_{SS} \tag{1}$$

Here $n_N(Dp)$ is the particle number size distribution and $Dp_{max}$ its upper limit, $CCN_{SS}$ is the measured CCN concentration at a given SS. Particle losses inside the CCN counter were accounted for by applying a size-dependent transmission



function (Rose et al., 2010) to particle number size distributions. With the obtained $Dp_{crit}$, the kappa hygroscopicity parameter ($\kappa_{CCN}$-value) was calculated according to Petters and Kreidenweis, 2007:

$$\kappa_{CCN} = \frac{4A^3}{27Dp_{crit}^3 S}$$     (2)

Where S is the saturation ratio, related to SS as S = SS/100+1. The above-described linear SS calibration curve was applied hereto. The constant A is defined as:

$$A = \frac{4\sigma_{a/w} M_w}{RT \rho_w}$$     (3)

Where $\sigma_{a/w} = 0.072 \, J \, m^{-2}$ is the surface tension of the air/water interface, T = 298.15 °C is the standard temperature
and $\rho_w$ is the density of water at temperature T, R the gas constant and $M_w$ is the molecular weight of water.

We found a discrepancy between the total particle number concentration measured by the CCN counter and the SMPS during the first ten days of the spring study, that we could not explain. We found that $CCN_{max}$ was significantly higher than the simultaneous number concentration of particles larger than 25 nm ($N_{25}$) measured with the SMPS. This could not be explained by either non-isokinetic sampling conditions in the duct from where the CCN sample air was drawn
(Supplementary material 2), or by decreased SMPS sample flow or other irregularities. We suspect the discrepancy to be caused by conditions in the sampling duct, possibly turbulence, as a change of the position of the CCN sampling port in the duct remedied the situation. The discrepancy was observed during 20–30 April. As the SMPS serves as a long-term monitoring device, we choose to accept it as our reference instrument. The ratio $CCN_{max}/N_{25}$ serves as a tool for assessing the behavior of the CCN counter against the SMPS measurements. The ratio was on average 1.614 in the period 20–30
April, whereas it was 0.939 for the rest of the CCN measurement period. We observed a strong linear correlation between $CCN_{max}$ and $N_{25}$ for the period after 30 April. By plotting $N_{25}$ as a function of $CCN_{max}$ we observed that $CCN_{max}$ increased by a constant factor during the period of discrepancy, and that there seemed to be no patterns in the offset. Thus, we have chosen to include this period of discrepancy, as it contains highly valuable measurements. We corrected all measured CCN concentrations within this period with a constant factor so that $CCN_{max}/N_{25}$ matches that of the rest of the
measurement period. A plot showing $N_{25}$ as a function of $CCN_{max}$ for both the uncorrected and corrected measurements before 30 April, and the remaining measurements after 30 April is included in Supplementary material 3 (Fig. S2).

## 2.4   Particle hygroscopic growth factor

Particle hygroscopic growth factor was measured with a Humidified Tandem Differential Mobility Analyzer (HTDMA)
Model 3100 from Brechtel (Lopez-Yglesias et al., 2014). The instrument measured HGF of particles with 30, 60, 120 and 240 nm dry diameter ($Dp_{dry}$) at 85 and 90 % RH. The HTDMA has a built-in aerosol dryer in the upstream classification unit. Due to the low ambient particle number concentrations in the high Arctic environment, the humidified size scans at each $Dp_{dry}$ were conducted over 10 min, also scans with increasing and decreasing DMA voltage were averaged. This procedure ensured sufficient particle counts in most instances. Data inversion was done by an Igor® algorithm based on
the principles in Stolzenburg and McMurry (2008) provided by Brechtel. As in the case of the CCN measurements, HGF measurements coinciding with polluted or faulty SMPS measurements were discarded. Moreover, maximum temperature changes of 1 °C h$^{-1}$ during a humidified scan and minimum total particle concentrations of 1 cm$^{-1}$ in the classified and humidified aerosol size spectra were used as quality control criteria.

The HTDMA system was fitted with an automatic atomizer setup, allowing for regular HGF control measurements of AS
aerosol. Control measurements of 100 nm AS particles at 85 and 90 % RH were conducted twice daily. Also, the consistency of the two particle sizing stages of the HTDMA was verified with size-certified polystyrene latex (PSL) spheres against a TSI 3080 SMPS system. We found that the set RH was reached within <2 % RH accuracy (Supplementary material 4). We used κ-Köhler theory to correct for these deviations (Petters and Kreidenweis, 2007). The $\kappa_{HTDMA}$-value can be inferred from HTDMA measurements by:



$$\frac{RH}{exp\left(\frac{A}{Dp_{dry} \cdot GF}\right)} = \frac{GF^3 - 1}{GF^3 - (1 - \kappa_{HTDMA})}$$
(4)

In this procedure, the actual operating RH, determined from AS control measurements, was used to calculate the $\kappa_{HTDMA}$-value. The HGF at 85 or 90 % RH was found by applying this $\kappa_{HTDMA}$-value to the formula and determining HGF at the specific RH by an iterative approach described in Supplementary material 5. Ideally, the $\kappa$-value of a multispecies aerosol can be determined by the $\kappa$-mixing rule:

$$\kappa = \sum \varepsilon_i \kappa_i$$
(5)

where $\varepsilon_i$ is the respective particle volume fraction of component i, and $\kappa_i$ is the $\kappa$-value for that component. This relationship can be used both for $\kappa$-values determined by CCN- or HGF measurements.

### 2.5 Determination of uncertainties

An iterative Monte Carlo approach was used to determine $Dp_{crit}$, $\kappa_{HTDMA}$ and the associated uncertainty of these parameters. The method is similar in concept to that used in Kristensen et al. (2016) and Herenz et al. (2018).

The counting process of the CCN counter was assumed to be Poisson. Hence, the standard deviation (SD) of the CCN concentration within the 5 min of data acquisition is related to the square root of total counts within that period. Even though the CCN concentration was generally low, the long data acquisition time resulted in counting SD rarely exceeding
1%. The uncertainty of SMPS sizing and concentration was assumed to be normally distributed. When determining $Dp_{crit}$, the SD of the particle diameter was estimated to be 2.5%, while a SD of 5% was estimated on the particle number concentration. The value of $Dp_{crit}$ was calculated according to Eq. (1) where the parameter values were drawn randomly from normal distributions centered at the given value with the above-stated standard deviations. Each calculation was iterated 1000 times, and the final $Dp_{crit}$ and its associated SD were determined by an automatic fit of a normal distribution
to the obtained set of results. An example histogram from this procedure is given in supplemental material 6 (Fig. S5).

To determine $\kappa_{HTDMA}$ a similar approach as for $Dp_{crit}$ was applied. The SD of RH found during the monitoring of the HTDMA RH was used, together with the same estimate of particle sizing uncertainty as above 2.5%. Again, fitting normal distributions to the obtained set of results yielded values of $\kappa_{HTDMA}$ with associated SD. The HGF at exactly 85 and 90% was then recalculated in accordance with Eq. (4), along with low and high estimates by subtracting from or adding the
SD to $\kappa_{HTDMA}$.

Meaningful uncertainty intervals could not be determined for $\kappa_{CCN}$ as Eq. (2) is highly sensitive to deviations in $Dp_{crit}$, resulting in skewed result distributions and unrealistic uncertainty intervals. Values of $\kappa_{CCN}$ are therefore reported without an uncertainty interval, but it is noted that these quantities inherently are associated with large uncertainties.

## 3 Results

### 3.1 Meteorology

Figures 2 and 3 show the evolution of meteorological parameters during the two field studies. We use UVA radiation as a proxy for solar radiation and gas-phase photo-oxidation of organic compounds, and thus new particle formation and growth. Precipitation measurements were not available for 4 – 27 May. A power outage caused measurements of
temperature, RH, radiation, wind speed, and pressure to fail from 25 August to 28 September. During this period, backup measurements of temperature, RH, wind speed, and pressure were available, however, a noticeable offset is seen for the pressure measurements.

During the first weeks of the spring study (Fig. 2a) temperatures were low, about -30 to -10 °C, with occasional snowfall. Later in the study, the temperature increased to just below the freezing point, and the composition of the concurrent





precipitation was probably a snow-rain mixture. RH ranged mostly between 70 and 90 % with a short decrease on 26
        April and occasional peaks later in the study. Time-lapse photography revealed that these peaks were associated with fog,
        and concurrent removal of accumulation mode particles was also observed. The entire spring study occurred during polar
        day (Fig. 2b). Correspondingly, UVA radiation was present, with an increasing trend during the study. Atmospheric
        pressure was high, especially during the first period until 13 May, but occasions with air pressure drops and high wind
speeds appeared several times in the latter half of this first study period.

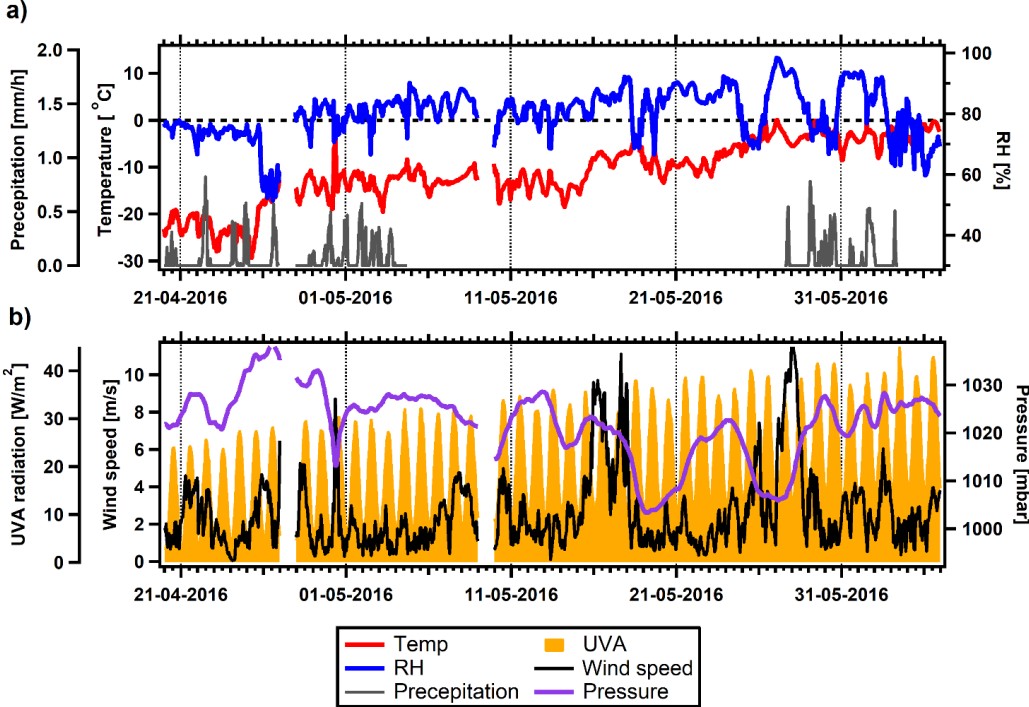

**Figure 2. Overview of meteorological conditions during the spring field study.** a) Temperature, RH and precipitation.
The dashed line indicates 0 °C. b) UVA radiation, wind speed and atmospheric pressure.

During the summer field study, temperatures were mostly above freezing in August (Fig. 3a). Thereafter they decreased
        to -10 °C towards the end of the summer study period. RH was more variable in this period compared to spring, ranging
        from 50 to 90% RH. The beginning of the summer study was still conducted during polar day, however, the UVA radiation
        was decreasing and almost disappeared at the end of this study period. This is expected, as polar sunset occurred about
        two weeks after the conclusion of the study. Atmospheric pressure was lower in summer compared to spring, with several
occasions of strong decreases of pressure and high wind speeds. Especially the period of 28 - 31 August, which featured
        higher wind speeds.

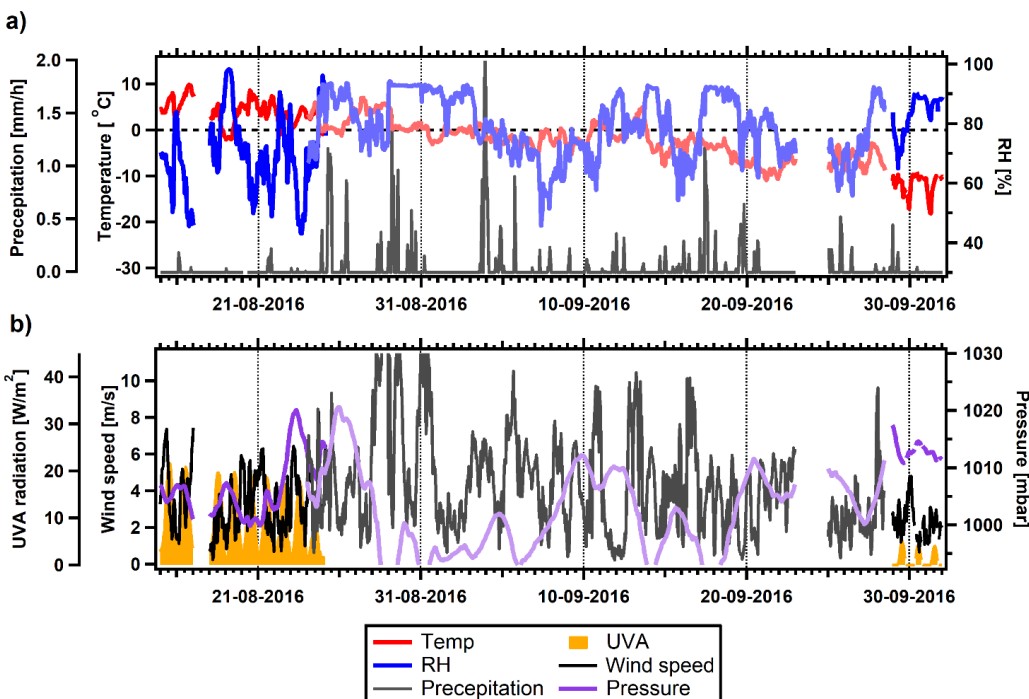

**Figure 3. Overview of meteorological conditions during the summer field study.** Lighter shaded lines are backup measurements. a) Temperature, RH and precipitation. The dashed line indicates 0 °C. b) UVA radiation, wind speed and atmospheric pressure.

### 3.2 Ambient particle number size distributions

The average ambient particle number size distributions during the two measurement periods shown in Fig. 4 are typical for Arctic conditions during spring and summer. During the first part of the spring period (20 April – 10 May), a distinct accumulation mode is visible at approximately 100-300 nm. This is the characteristic Arctic haze mode, which predominantly consists of anthropogenic aerosols that are built up during winter and spring. In the second part of the spring period (11 May – 2 June), the haze mode largely disappeared and in turn, a nucleation mode appeared.

In the first part of the summer period (15 August – 10 September), the nucleation mode has grown compared to the latter part of the spring study. An Aitken mode at around 20-30 nm can also be identified. This signifies that secondarily formed particles succeeded to grow into the Aitken mode. In the latter part of the summer period, the number of freshly formed particles decreased, and as an accumulation mode has not yet significantly developed, the total particle number concentration becomes low. The large uncertainty bars on the graphs show that deviations from the average distributions are quite large. For example, new particle formation still happened in the latter part of September, but on fewer occasions.



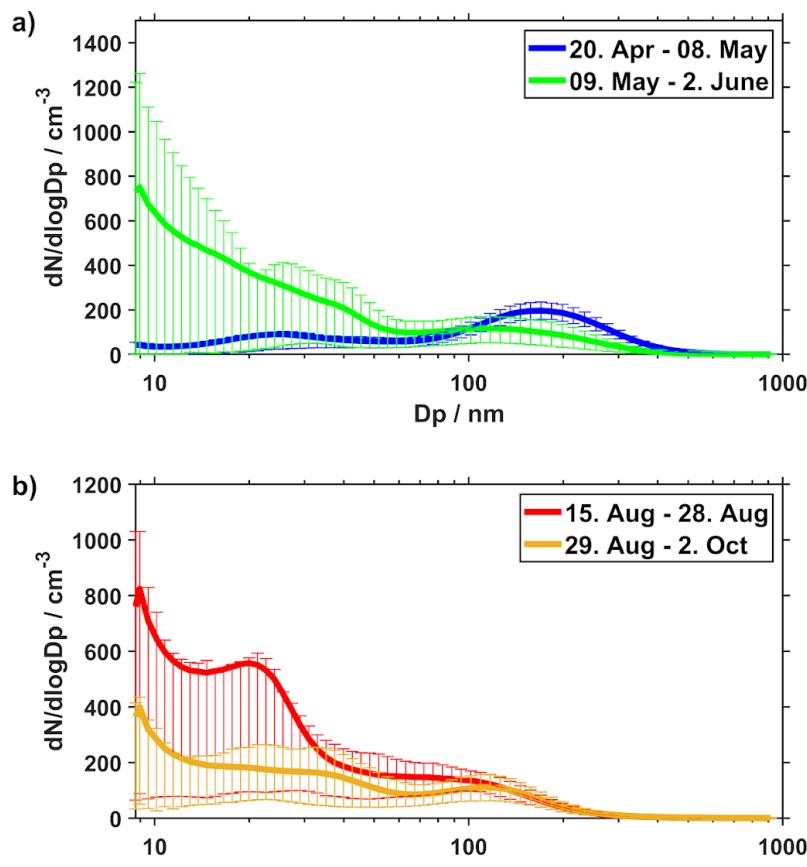

**Figure 4. Mean particle number size distributions** during a) the spring measurement period and b) the summer measurement period. Error bars show 25th/75th percentiles in each SMPS size bin over the respective period.

### 3.3 Cloud condensation nuclei

The time series of CCN concentration, $Dp_{crit}$ and $\kappa_{CCN}$ for 0.1–1.0 % SS are shown for the spring period in Fig. 5, and the summer period in Fig. 6. In less than 0.1 % of all measurements at 1.0 % SS, a $Dp_{crit}$ <25 nm can be identified. The average $Dp_{crit}$ at ~1.8 % SS was 33.7 nm, while the average activation fraction ($CCN_{max}/N_{25}$), as mentioned in Sect. 2.3, was 0.939, which gives credibility to our CCN measurements.

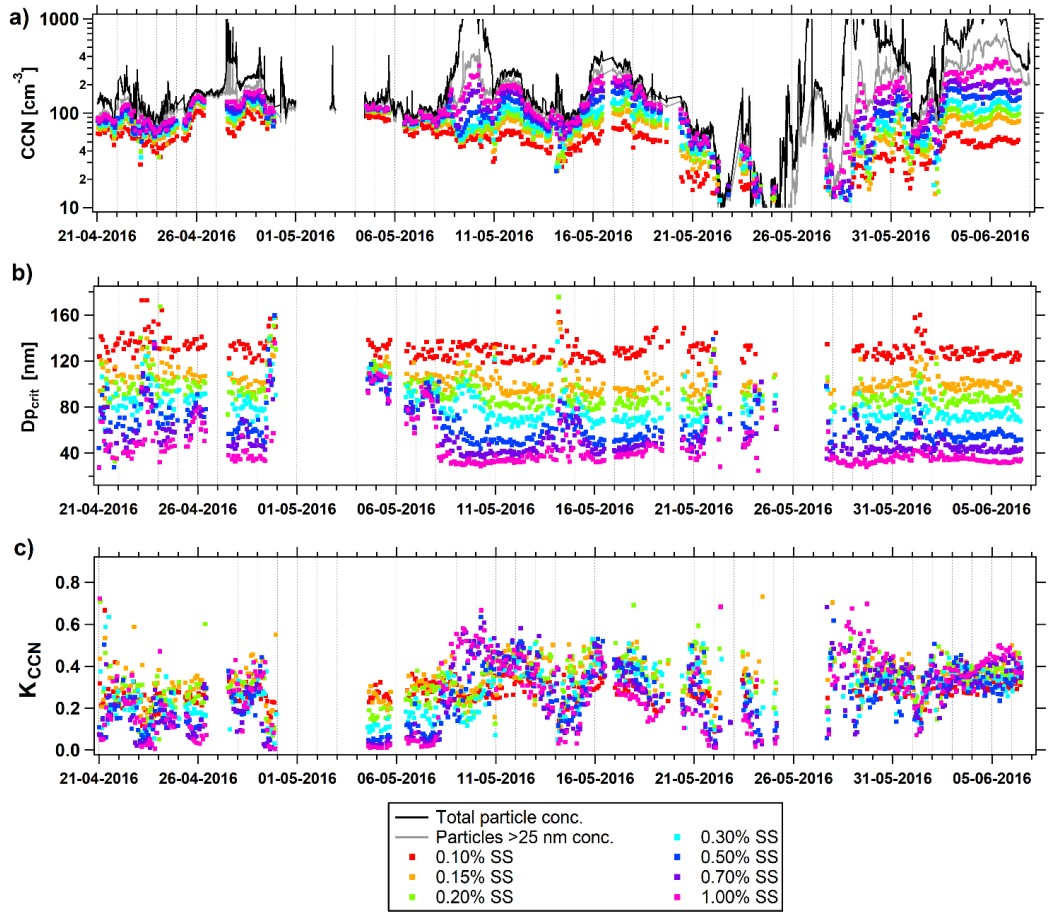


**Figure 5. CCN properties at various SSs during the spring measurement period.** a) particle and CCN number concentration, b) critical activation diameter ($Dp_{crit}$), and c) $\kappa_{CCN}$-value. Black line: total particle number concentration inferred from SMPS measurements ($N_{tot}$), gray line: number concentration of particles >25 nm ($N_{25}$), colored symbols: CCN measurements at 0.1–1.0 % SS.


Up to 8 May, CCN concentrations at varying SSs follow a similar pattern and differences between the values are relatively small (Fig. 5). From then and ongoing, differences for measured CCN concentrations are much larger across all SSs. This indicates that before May 8 a quite homogenous aerosol was observed, where even the activation of particles with smaller diameters at higher SS did not appreciably change the total number of activated particles, because accumulation mode particles dominated during this period. In contrast, during the latter period (May 9–June 2), applied SS played a more critical role with respect to the total number of activated particles. While there was little difference between $CCN_{0.1}$ and $CCN_{1.0}$ in the beginning of the spring field study, the difference between these two parameters was larger towards its end (Fig. 5). A subsequent increase in SS led to more activated particles in the Aitken mode range, identifying the importance of Aitken mode aerosol during this period. The days around 8 May seem to mark a transition from the typical Arctic haze period in late winter/early spring to the late spring/summer regime where transport of air masses from mid-latitudes to







the high Arctic is minimized. This is supported by a significant decrease in non-sea salt sulfate concentrations measured at Villum during the same period (see Skov et al., 2017 for further details and filter pack data from EBAS database).

It is interesting to note, that while $Dp_{crit}$ varies at the higher supersaturations, it appears almost constant at 0.1% and 0.15 % SS (Figure 5b). Correspondingly, $Dp_{crit}$ and $\kappa_{CCN}$-values at 0.1 and 0.15% SS show the smallest temporal variation
throughout the spring field study, from all the listed supersaturations (Table 2). This behavior could be explained by a relatively unchanged chemical composition of the larger accumulation mode particles that activated at these lower SSs. A general trend is that the spread of CCN concentrations seen at different SSs increased during the second part of the spring field study (Figure 5a). Further CCN property statistics are stated in Tables 2 and 3.

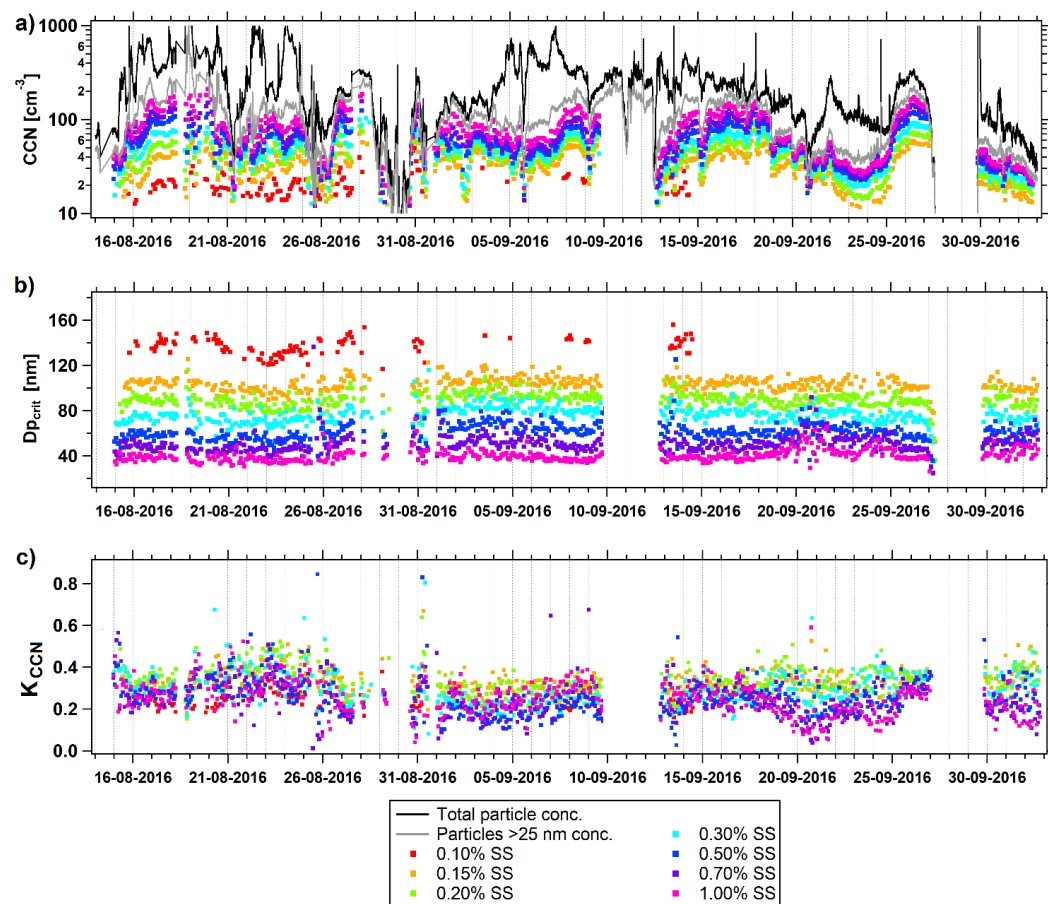


**Figure 6. CCN properties at various SSs during the summer measurement period.** a) Particle and CCN number concentration, b) critical activation diameter ($Dp_{crit}$), and c) $\kappa_{CCN}$ value. Black line: total particle number concentration inferred from SMPS measurements ($N_{tot}$), gray line: number concentration of particle >25 nm ($N_{25}$), colored symbols: CCN measurements at 0.1-1.0% SS.



The summer field study is characterized by relatively small variations of the observed critical diameters and corresponding kappa values. This finding indicates a relatively stable and homogeneous chemical composition over the whole summer period of observations, which is not surprising as the high Arctic summer aerosol is dominated by local and regional emissions. Simultaneously, relatively high variations are observed in the number of total activated particles, with all SSs
showing similar patterns. An explanation for this finding might be relatively stable emission sources and processing pathways, with the emission source strength varying, which results in a stable shape of the particle number size distribution with varying total particle number concentration. As observed during the second period of the spring field study, differences in activated particle numbers at different applied SSs are still relatively large. This indicates a dominating Aitken mode where higher SSs activate larger fractions (lower $Dp_{crit}$) of this aerosol mode, which
predominantly originates from local and regional emissions.

**Table 2**. **Statistics of CCN measurements for the spring field study.** The uncertainty values stated with CCN mean, CCN median, $Dp_{crit}$ mean, and $Dp_{crit}$ median are the respective statistics of the associated uncertainty, found in the fitting procedure. The values of SD are the standard deviations of the parameter distributions over the temporal evolution of the
study. *signifies a significant difference between spring and summer field studies (Wilcoxon rank sum test, $p < 0.05$). CCN measurements for the spring field study were obtained from 20 April to 6 June.

| Spring | 0.1% SS | 0.15% SS | 0.2% SS | 0.3% SS | 0.5% SS | 0.7% SS | 1.0% SS |
|---|---|---|---|---|---|---|---|
| CCN (uncert) mean [cm⁻³] | 53.7* (±0.5) | 69.3* (±0.5) | 76.5* (±0.5) | 85.3* (±0.6) | 98.4* (±0.6) | 112.7* (±0.6) | 133.8* (±0.7) |
| CCN (uncert) median [cm⁻³] | 53.3* (±0.5) | 71.3* (±0.5) | 78.6* (±0.6) | 85.1* (±0.6) | 94.9* (±0.6) | 108.3* (±0.7) | 116.6* (±0.7) |
| CCN SD [cm⁻³] | 20.2 | 25.6 | 28.0 | 32.7 | 41.7 | 53.9 | 75.3 |
| $Dp_{crit}$ (uncert) mean [nm] | 130.2* (±21.4) | 102.9* (±25.3) | 92.2 (±26.8) | 80.2 (±25.8) | 64.2 (±22.0) | 54.4* (±19.2) | 44.8* (±16.6) |
| $Dp_{crit}$ (uncert) median [nm] | 129.2* (±19.2) | 100.7* (±25.0) | 88.9 (±25.9) | 75.9 (±25.3) | 58.7 (±21.4) | 46.7* (±18.0) | 37.3* (±16.0) |
| $Dp_{crit}$ SD [nm] | 9.5 | 10.9 | 13.1 | 14.1 | 16.8 | 19.1 | 19.2 |
| $\kappa_{CCN}$ mean | 0.28* | 0.35* | 0.33 | 0.29 | 0.27 | 0.27* | 0.30* |
| $\kappa_{CCN}$ median | 0.28* | 0.35* | 0.34 | 0.29 | 0.27 | 0.29* | 0.30* |
| $\kappa_{CCN}$ SD | 0.06 | 0.10 | 0.11 | 0.12 | 0.15 | 0.16 | 0.19 |



**Table 3**. **Statistics of CCN measurements for the summer field study.** The uncertainty values stated with CCN mean, CCN median, $Dp_{crit}$ mean, and $Dp_{crit}$ median are the respective statistics of the associated uncertainty, found in the fitting procedure. The values of SD are the standard deviations of the parameter distributions over the temporal evolution of the study. *signifies a significant difference between spring and summer field studies (Wilcoxon rank-sum test, $p<0.05$). CCN measurements for the summer field study were obtained from 15 August to 2 October.

| Summer | 0.1% SS | 0.15% SS | 0.2% SS | 0.3% SS | 0.5% SS | 0.7% SS | 1.0% SS |
|---|---|---|---|---|---|---|---|
| CCN (uncert) mean [cm⁻³] | 20.8* (±0.3) | 33.2* (±0.4) | 39.6* (±0.4) | 47.6* (±0.4) | 56.6* (±0.5) | 65.6* (±0.5) | 79.5* (±0.5) |
| CCN (uncert) median [cm⁻³] | 19.4* (±0.3) | 32.5* (±0.4) | 39.2* (±0.4) | 46.2* (±0.4) | 53.0* (±0.5) | 59.8* (±0.5) | 73.9* (±0.5) |
| CCN SD [cm⁻³] | 6.2 | 12.4 | 15.7 | 19.4 | 24.9 | 31.1 | 40.0 |
| $Dp_{crit}$ (uncert) mean [nm] | 137.1* (±10.0) | 103.3* (±19.1) | 89.7 (±20.7) | 76.7 (±22.0) | 61.6 (±20.7) | 51.3* (±18.6) | 40.5* (±16.3) |
| $Dp_{crit}$ (uncert) median [nm] | 138.7* (±6.4) | 103.6* (±15.8) | 90.0 (±18.0) | 76.6 (±20.2) | 60.7 (±19.8) | 49.7* (±16.9) | 39.3* (±14.2) |
| $Dp_{crit}$ SD [nm] | 7.78 | 6.51 | 6.7 | 8.2 | 9.15 | 9.12 | 5.55 |
| $\kappa_{CCN}$ mean | 0.23* | 0.34* | 0.35 | 0.30 | 0.25 | 0.24* | 0.27* |
| $\kappa_{CCN}$ median | 0.22* | 0.32* | 0.33 | 0.28 | 0.24 | 0.23* | 0.27* |
| $\kappa_{CCN}$ SD | 0.04 | 0.13 | 0.23 | 0.15 | 0.11 | 0.12 | 0.10 |


The difference between the mean and median values in Tables 2 and 3 illustrates to which extent extreme values influence the respective parameters. Symmetric distributions have equal values of mean and median, hence differences are caused by skewed distributions. The mean CCN concentration was influenced by low concentrations during the spring study for 0.15 - 0.2 % SS, while higher concentrations elevated the mean above the median for 0.5 - 1.0 % SS. Elevated mean CCN
concentrations were observed for all supersaturations during the summer field study, although mean and median values were within the uncertainty range for 0.15–0.3 % SS. Given the low aerosol concentrations during the summer, any perturbation in the CCN burden will elevate the mean. The stated standard deviation for CCN, $Dp_{crit}$ and $\kappa_{CCN}$ signifies the temporal variation of these properties. For $Dp_{crit}$, the standard deviation was smaller for the summer study compared to spring. Meanwhile, $Dp_{crit}$ median was consistently larger during summer compared to spring. Median and mean $\kappa_{CCN}$
was consistently larger in spring compared to summer. The standard deviation of $\kappa_{CCN}$ was larger at 0.15 - 0.3 % SS during summer, but larger at 0.5 - 1.0 % SS during spring. This indicates that particles in the accumulation mode range, with $Dp_{crit}$ of about 70 - 100 nm, had a more variable chemical composition during summer, while Aitken mode particles, with $Dp_{crit}$ ~35 - 60 nm, had a more variable chemical composition during the spring measurement period. Thus, it can be concluded that sources of accumulation mode particles were more stable and consistent in spring, while they were more
variable and diverse in summer. Conversely, sources of Aitken mode particles were more variable in spring, and more stable and confined in summer.

Except for the total CCN measured at different supersaturations, significant differences for the retrieved hygroscopicity parameters $Dp_{crit}$ and $\kappa_{CCN}$ (mean and median) were observed only for the two lowest and the two highest supersaturations. As low supersaturations - based on our cumulative approach to retrieve $Dp_{crit}$ and therewith $\kappa_{CCN}$ - are mostly affected by
the larger particles, we conclude that there are substantial differences in the chemical composition of the larger particles between the two seasons. As high supersaturations are mostly affected by the whole size distribution including the very





small particles, we can also conclude that there are substantial differences in the chemical composition of the smallest particles between the two seasons. This is supported by the chemical composition of nucleating vapors at Villum (Beck et al., 2021), which showed that during the spring iodic acid is the main precursor vapor while in the summer sulfuric acid and ammonium become the dominant precursor species. These differences between the seasons for different sizes are somewhat expected as the spring particle number size distribution is mostly affected by processed and larger particles compared to the summer where freshly and locally formed particles affect the particle number size distribution.

We deliberately choose to report the total mean and median values of the entire set of calculated uncertainties, rather than combining the uncertainties by dividing the standard deviation by the square root of the number of samples. This is because the individual sample points cannot be assumed to sample the same quantity. As the aerosol changed with time, the CCN measurement process sampled a changing variable. Hence, the individual measurements did not sample identical probability distribution functions, and the traditional approach for finding the error of the mean is not valid.

CCN concentration and $\kappa_{CCN}$ at 0.3% SS were coupled with 96 h air mass back trajectories and sea ice maps, in order to investigate possible sources of particular CCN active aerosols. The trajectories were based on meteorological data from the NCEP/NCAR Reanalysis Project (Kalnay et al., 1996) and calculated with HYSPLIT 4 (arrival height 100 m a.s.l. at 00:00, 06:00, 12:00 and 18:00h). Maps of sea-ice concentration were taken from the National Sea & Ice Data Center (https://nsidc.org/data/seaice_index/archives/). Besides weak indications that higher CCN concentrations reached Villum from the Greenlandic west coast compared to the central Arctic Ocean or Greenlandic east coast during spring, and that the lowest CCN concentrations reached Villum from the Canadian Archipelago during summer, no robust correlations could be determined.

During late summer, on 31 Aug, an increased $\kappa_{CCN}$ of 0.79 at 0.3% occurred with high wind speeds from the direction of the unfrozen fjord, linking this measurement with hygroscopic sea spray aerosols (Martensson et al., 2003; de Leeuw et al., 2011). This is further supported by the concurrent ambient number size distributions (Fig. S6), which showed a bimodal appearance that is indicative of sea spray aerosol (Quinn et al., 2017).

### 3.4 Particle hygroscopic growth factor

To compensate for RH deviations inside the HTDMA, we determined the $\kappa_{HTDMA}$ according to the method described in Sect. 2.4 and recalculated the HGF at exact values of 85 and 90 % RH. The time series of these recalculated particle hygroscopic growth factors are presented for the spring study in Fig. 7 and the summer study in Fig. 8. In only very rare cases did we observe bimodal HGF distributions (2.4 % in spring, 0.2 % in summer), signifying a predominantly internally mixed aerosol for both study periods. These instances are neglected, and an internal mixture is assumed at all times.

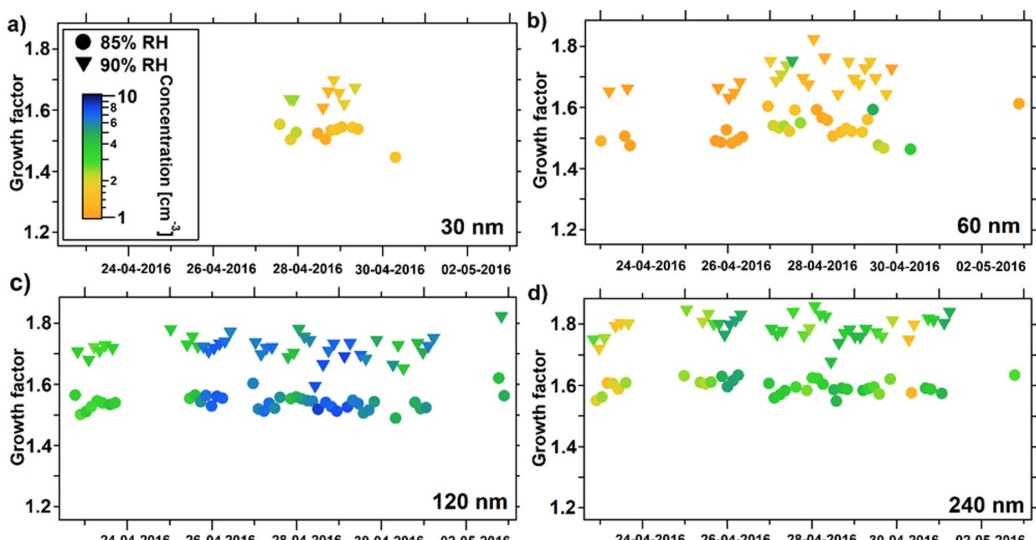

**Figure 7. Particle hygroscopic growth factors during the spring study.** a) Growth factors at 30 nm $Dp_{dry}$, b) at 60 nm $Dp_{dry}$, c) at 120 nm $Dp_{dry}$ and, d) at 240 nm $Dp_{dry}$. Dots represent values at 85% RH and triangles at 90% RH. The color indicates the raw particle number concentration measured at that certain $Dp_{dry}$, downstream of the first and upstream of the second DMA.

Figure 7a contains fewer data points than the other panels. This was caused by the often too low particle concentrations observed at 30 nm $Dp_{dry}$ during the spring study period. The HGFs at 85 and 90 % followed similar trends for all selected dry particle diameters. Generally, hygroscopic growth factors did not vary greatly during the spring study but it appears that variations were slightly stronger at 90% compared to 85 % RH as expected. There was a general increase of HGF with increasing $Dp_{dry}$, which is further shown in Table 4 below. In general, relatively high HGFs were found for the spring campaign assuming ammonium bisulfate being a major compound of the observed aerosol. While $\kappa_{HTDMA}$ mean values range between 0.54 and 0.60 at RH = 85 % and between 0.46 and 0.51 at RH = 90 % (Table 4), for comparison a theoretical value for ammonium sulfate is stated as 0.53 (Petters and Kreidenweiss et al., 2007). The occurrence of partly unneutralized aerosols has been found earlier by Nielsen et al. (2019) during a spring study in 2015 using a Soot Particle - Aerosol Mass Spectrometer (SP-AMS), which is in accordance with our findings here. The overall HGF timeline shows a relatively stable aerosol, especially for accumulation mode particles with respect to subsaturated hygroscopic growth which might be related to a transport period including predominantly haze aerosol with only little variation. This finding can be confirmed by the small dependence of total CCN on different SSs implying that the majority of the particles are linked to an accumulation mode. The overall HGF timeline shows a relatively stable aerosol, especially for accumulation mode particles with respect to subsaturated hygroscopic growth which might be related to a transport period including predominantly haze aerosol with only little variation. This finding can be confirmed by the small dependence of total CCN on different SSs implying that the majority of the particles are linked to an accumulation mode.

However, fluctuations of individual values in HGF around the mean values can well be explained by a variation in the degree of neutralization of the aerosol. The distribution of HGF over the size regime as well as the corresponding calculated $\kappa_{HTDMA}$ imply only little changes of the chemical composition over the size range of the Arctic aerosol investigated at Villum during this spring study.



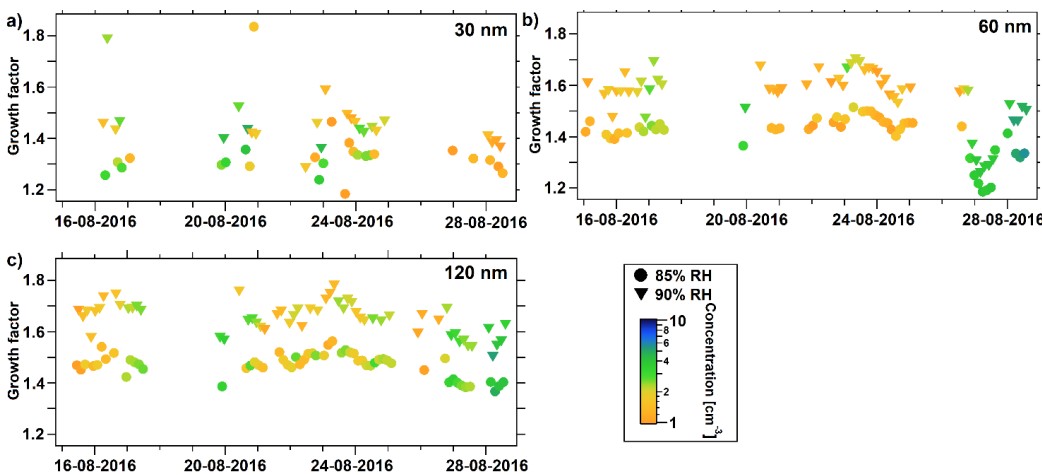

**Figure 8. Particle hygroscopic growth factors during the summer study.** a) Growth factor at 30 nm $Dp_{dry}$, b) at 60 nm $Dp_{dry}$, and c) at 120 nm $Dp_{dry}$. Dots represent values at 85% RH and triangles at 90% RH. The color indicates the raw
particle number concentration measured at that certain $Dp_{dry}$, downstream of the first and upstream of the second DMA.

Stronger variations of HGF were observed during the summer study period compared to the spring study. Also, a larger
number of measurements were carried out at 30 nm $Dp_{dry}$, but no valid measurements could be obtained at 240 nm. In
general, the pattern of HGF at 60 and 120 nm appear to covary, while measurements at 30nm $Dp_{dry}$ are inconsistent with
those at larger sizes indicating a different chemical composition for the smallest particle size range. As an overall finding,
HGF during the summer study was lower compared to the spring study. These differences are statistically significant for
the mean and median values measured at RH = 85% for all sizes and at RH = 90 % for 30 nm and 60 nm $Dp_{dry}$.
Correspondingly,  $\kappa_{HTDMA}$ mean values ranging from 0.37 to 0.45 (RH = 85 %) and from 0.36 to 0.45 (RH = 90 %), which
no longer reflect a pure chemical inorganic composition of ammonium bisulfate, but rather imply that the observed aerosol
might be composed, to a substantial extent, of organic compounds mixed with inorganics species. This finding reflects
the overall theory that the summer Arctic aerosol is predominantly of local and regional origin with microbial activity in
marginal ice zones and phytoplankton sea surface emissions delivering the inorganic as well as the organic ingredients
for particle growth. Organic aerosol precursors, which partly condense onto existing aerosol, might be an explanation for
this finding by lowering the overall hygroscopicity of the observed particles leading to lower $\kappa_{HTDMA}$ mean values. Low-
level transport of anthropogenic pollution is to a large extent very limited during this time of the year. Statistics of
measured particle hygroscopic growth factors and associated $\kappa_{HTDMA}$-values for both study periods are listed in Tables 4
and 5.





**Table 4. Statistics of HTDMA measurements during the spring study.** The stated uncertainties are the respective statistics of the associated uncertainty retrieved from the fitting procedure as described in Sect. 2.5. The values of SD are the standard deviations of the parameter distributions over the temporal evolution of the studies. *signifies a significant difference between spring and summer field studies (Wilcoxon rank-sum test, $p<0.05$). HTDMA measurements for the spring field study were obtained from 22 April to 2 May.

| Spring | 30 nm | 60 nm | 120 nm | 240 nm |
|---|---|---|---|---|
| HGF 85% RH (low-upper) mean | 1.53* (1.48-1.58) | 1.55* (1.49-1.61) | 1.56* (1.50-1.62) | 1.62 (1.55-1.68) |
| HGF 85% RH (low-upper) median | 1.55* (1.50-1.60) | 1.54* (1.49-1.60) | 1.56* (1.50-1.62) | 1.62 (1.55-1.68) |
| HGF 85% RH SD | 0.05 | 0.04 | 0.03 | 0.03 |
| HGF 90% RH (low-upper) mean | 1.60* (1.51-1.68) | 1.69* (1.58-1.79) | 1.68 (1.56-1.79) | 1.75 (1.62-1.87) |
| HGF 90% RH (low-upper) median | 1.61* (1.52-1.69) | 1.66* (1.55-1.76) | 1.69 (1.57-1.79) | 1.76 (1.62-1.87) |
| HGF 90% RH SD | 0.06 | 0.09 | 0.04 | 0.04 |
| $\kappa_{HTDMA}$ 85% RH (uncert) mean | 0.60* (±0.08) | 0.56* (±0.08) | 0.54* (±0.08) | 0.59 (±0.09) |
| $\kappa_{HTDMA}$ 85% RH (uncert) median | 0.63* (±0.08) | 0.55* (±0.08) | 0.53* (±0.08) | 0.59 (±0.09) |
| $\kappa_{HTDMA}$ 85% RH SD | 0.07 | 0.06 | 0.04 | 0.02 |
| $\kappa_{HTDMA}$ 90% RH (uncert) mean | 0.50* (±0.10) | 0.52* (±0.12) | 0.46 (±0.11) | 0.51 (±0.13) |
| $\kappa_{HTDMA}$ 90% RH (uncert) median | 0.51* (±0.10) | 0.48* (±0.11) | 0.47 ±(±0.11) | 0.52 (±0.13) |
| $\kappa_{HTDMA}$ 90% RH SD | 0.07 | 0.11 | 0.04 | 0.04 |





**Table 5. Statistics of HTDMA measurements during the summer field study.** The stated uncertainties are the respective statistics of the associated uncertainty retrieved from the fitting procedure as described in Sect. 2.5. The values of SD are the standard deviations of the parameter distributions over the temporal evolution of the studies. *signifies a significant difference between spring and summer field studies (Wilcoxon rank-sum test, $p<0.05$), there are no measurements at 240 nm $Dp_{dry}$ in the summer field study. HTDMA measurements for the summer field study were obtained from 15 August to 28 August.

| Summer | 30 nm | 60 nm | 120 nm | 240 nm |
|---|---|---|---|---|
| HGF 85% RH (low-upper) mean | 1.36* (1.33-1.39) | 1.43* (1.40-1.46) | 1.50* (1.46-1.53) | n.a. |
| HGF 85% RH (low-upper) median | 1.34* (1.31-1.37) | 1.45* (1.42-1.47) | 1.51* (1.46-1.55) | n.a. |
| HGF 85% RH SD | 0.12 | 0.09 | 0.05 | n.a. |
| HGF 90% RH (low-upper) mean | 1.47* (1.41-1.52) | 1.58* (1.50-1.66) | 1.67 (1.58-1.75) | n.a. |
| HGF 90% RH (low-upper) median | 1.45* (1.39-1.51) | 1.61* (1.52-1.68) | 1.69 (1.59-1.77) | n.a. |
| HGF 90% RH SD | 0.09 | 0.11 | 0.10 | n.a. |
| $\kappa_{HTDMA}$ 85% RH (uncert) mean | 0.37* (±0.04) | 0.40* (±0.04) | 0.45* (±0.05) | n.a. |
| $\kappa_{HTDMA}$ 85% RH (uncert) median | 0.34* (±0.04) | 0.42* (±0.05) | 0.46* (±0.05) | n.a. |
| $\kappa_{HTDMA}$ 85% RH SD | 0.20 | 0.10 | 0.07 | n.a. |
| $\kappa_{HTDMA}$ 90% RH (uncert) mean | 0.36* (±0.06) | 0.41* (±0.08) | 0.45 (±0.09) | n.a. |
| $\kappa_{HTDMA}$ 90% RH (uncert) median | 0.34* (±0.06) | 0.43* (±0.08) | 0.47 (±0.09) | n.a. |
| $\kappa_{HTDMA}$ 90% RH SD | 0.10 | 0.10 | 0.08 | n.a. |

The HGF low and upper values in Tables 4 and 5 were derived by either subtracting or adding the uncertainty of $\kappa_{HTDMA}$ in the recalculation procedure, as described in Sect. 2.5. The uncertainty of $\kappa_{HTDMA}$, in turn, is the standard deviation of the normal distribution fit applied to the iteratively obtained set of $\kappa_{HTDMA}$-values. As noted for the CCN dataset, it cannot be assumed that the HTDMA measurements probe identical probability density functions, thus the combination of the standard deviation to a standard error is not a valid procedure. Again, here we instead state the mean and median values of the $\kappa_{HTDMA}$ and the corresponding uncertainties.

In summary, we can see that for all smaller diameters (30 nm and 60 nm $Dp_{dry}$) the HGF values (mean and median) at RH 85 and RH 90 % are significantly different between the two field studies. This is also valid for the retrieved $\kappa_{HTDMA}$ values. The two periods showed no significant difference at 120 nm $Dp_{dry}$ at 90 % RH. Similarly, $\kappa_{HTDMA}$ had a larger temporal variability in the summer period. Especially $\kappa_{HTDMA}$ at 30 nm had a comparably high temporal variability in summer. This indicated that as the hygroscopicity of 30 nm particles was on average lower during summer compared to spring, these



particles had a more variable hygroscopicity, and correspondingly a more variable chemical composition. As these particles are expected to originate from nucleation processes in the region, this indicates that various processes and thus different chemical compounds were responsible for the aerosol formation and/or further growth processes of these 30 nm particles during the summer study period (Beck et al., 2021).


### 3.5 Comparison of CCN and HTDMA hygroscopicity

The comparison of $\kappa$-values determined simultaneously (within 2 h) by HTDMA and CCN measurements, revealed differing patterns between the two study periods. The differing pattern was observed at all SSs, but here results from 0.3% SS and both 85% and 90% RH are shown, as this illustration presents the clearest picture. Spring and summer $\kappa_{HTDMA}$

versus $\kappa_{CCN}$ datasets are displayed in Fig. 9. In general, only weak correlations were found between $\kappa_{HTDMA}$ and $\kappa_{CCN}$. Earlier studies also found different values between the two techniques (e.g. Petters and Kreidenweis, 2007; Rastak et al., 2017; Rosati et al., 2020).

Particles composed of partly neutralized sulfates or sulfuric acid are expected to exhibit similar hygroscopic behavior compared to that observed for the spring measurements (Petters and Kreidenweis, 2007; Biskos et al., 2009), i.e.,

comparably higher hygroscopic growth at subsaturated conditions and lower CCN activity at supersaturation. This corresponds to other findings from Villum, where the haze aerosol was found to be acidic, containing a large fraction of non-neutralized sulfates (Nielsen et al., 2019). However, as the $\kappa$-values of un-neutralized sulfates are quite high (e.g. $\kappa_{HTDMA}(H_2SO_4) = 1.19$, $\kappa_{CCN}(H_2SO_4) = 0.90$), additional less hygroscopic species, e.g. biogenic or anthropogenic organics, other inorganic components, and/or black carbon, need to be a significant fraction of the observed internally mixed aerosol

as well to explain our observations.

During the summer period, it appears that the aerosol was less acidic. Ammonium sulfate exhibits lower $\kappa$-values compared to sulfuric acid and gives a better agreement between the two techniques ($\kappa_{HTDMA}(AS) = 0.53$, $\kappa_{CCN}(AS) = 0.61$). Basically, a ratio larger than 1 was found between median $\kappa_{HTDMA}$ and $\kappa_{CCN}$ for both seasons in our study. Reasons why an increase for median $\kappa_{CCN}$ was found from spring to summer compared to a decrease for median $\kappa_{HTDMA}$ for the

same period in our study remain unresolved and need further investigation, likely requiring a longer dataset. A possible explanation is stated in the following paragraph.



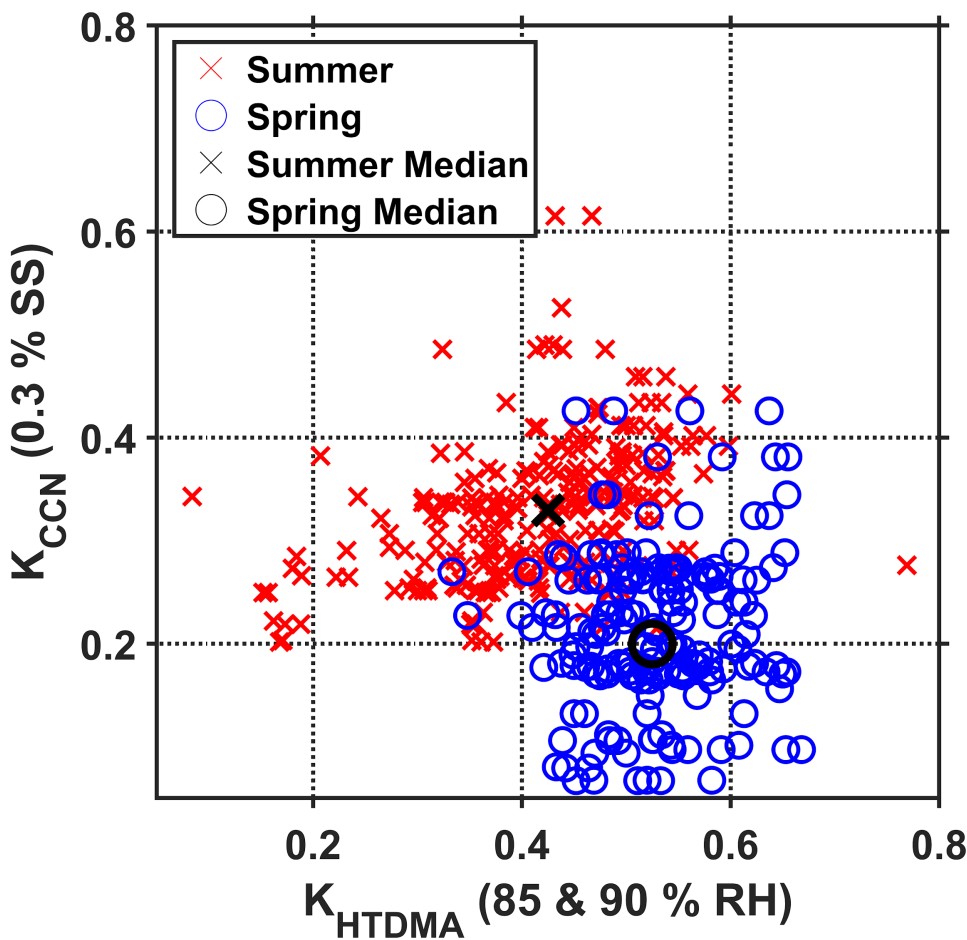

**Figure 9. Correlation of $\kappa_{HTDMA}$ and $\kappa_{CCN}$.** The $\kappa_{HTDMA}$-values include measurements at both 85 and 90% RH, of 30,
60, 120 and 240 nm $Dp_{dry}$. $\kappa_{CCN}$ was recorded at 0.3% SS. Blue circles are spring measurements, red crosses are summer
measurements. The black circle (cross) indicates the median for $\kappa_{HTDMA}$ and $\kappa_{CCN}$ during the summer (spring) field study.
This Figure is adopted from the supplement material in Lange et al., 2019.

The full-year study from Svalbard by Silvergren et al. 2014 shows an increase in $\kappa_{CCN}$ from April to August, from about
0.39 to 0.54, and an increase in $\kappa_{HTDMA}$ from about 0.38 to 0.52 (both ranges originate from Figure 9 in the referred study).
While we observe a corresponding increase in $\kappa_{CCN}$, we also observe a general decrease in $\kappa_{HTDMA}$ over the corresponding
period. This difference might be caused by a smaller relative contribution of sea salt components to the aerosol
composition at Villum, compared to Svalbard. Also, a smaller contribution from DMS-derived sulfate aerosol is expected
at Villum compared to Svalbard, based on the larger distance to open waters at Villum during summer. Marine microgel
aerosols have been described to feature efficient CCN properties while showing low hygroscopic growth at subsaturated
conditions (Ovadnevaite et al., 2011; Leck and Svensson, 2015; Hodas et al., 2016). As the coastline in the immediate
proximity of Villum was ice-free during the summer study period, and the Arctic pack ice extent was close to a minimum



during August, areas of open water and open pack ice were available as potential sources for marine aerosols and gels from the sea surface microlayer. However, we only see a very limited indication of these properties in our data.


## 4    Discussion and outlook

Our measurements show that aerosols of the Arctic atmosphere demonstrate relatively high CCN activity and hygroscopic growth at subsaturated conditions. This is noteworthy as organic compounds make up a significant mass fraction (Leck et al., 2002; Chang et al., 2011; Lange et al., 2018; Nielsen et al. 2019) in Arctic aerosols. Correspondingly, the organic
mass fraction must be assumed to contribute significantly to the hygroscopicity of the observed aerosol at Villum. For the spring and the summer measurement period, we determined average $\kappa_{CCN}$-values of 0.27–0.35 and 0.23–0.35, respectively. These values are mainly the result of the combined hygroscopicity of the main aerosol sulfate- and organic components. Nielsen et al. (2019) determined the organic mass component in $PM_1$ to be on average 24% over the period February–May at Villum. However, the organic component of the aerosol mass in the Arctic is highly size-dependent in
the particle size range probed both by the CCN counter and HTDMA in this study (Croft et al., 2019). Despite the relatively high $\kappa_{CCN}$-values, average CCN concentrations were quite low, especially during summer. For supersaturations ≤0.3 %, the spring period featured average CCN concentrations of 53.7–85.3 cm$^{-3}$ while average concentrations of 20.8–47.6 cm$^{-3}$ were observed during the summer period. For SS ≥0.5 %, CCN concentrations in spring were 98.4–133.8 cm$^{-3}$ while CCN concentrations of 56.6–79.5 cm$^{-3}$ were observed in summer. This shows that even when frequent NPF events
take place in summer, the atmosphere only contains only few potential CCN, and increases in anthropogenic emissions, either by industrial activities or increased ship traffic may have a correspondingly large impact.

The CCN properties determined in this study compare, to a varying extent, with previous findings from other parts of the Arctic. Our mean CCN concentration at 0.7 % SS of 112.7 and 65.6 cm$^{-3}$ in spring and summer, respectively, are somewhat higher than those of 14–47 cm$^{-3}$ determined in the central Arctic Ocean by Martin et al. (2011). Oppositely,
the summer median CCN concentration at 0.55 % SS of 247 cm$^{-3}$ determined by aircraft in Arctic air masses by Lathem et al. (2013) is significantly higher than the median CCN concentration at 0.50 % SS of 53.0–95.0 cm$^{-3}$ that was found during both our field studies. Martin et al. (2011) and Lathem et al. (2013) determined $\kappa_{CCN}$ (mean ± SD) to be 0.33 ± 0.13  and 0.32 ± 0.21, respectively, which compares well to our study even if they are slightly higher than our total field study (mean ± SD) $\kappa_{CCN}$ of 0.30 ± 0.21. A study by Jung et al. (2018), evaluated a several-year CCN dataset from Zeppelin
Mountain, Svalbard. They found CCN concentrations at 0.2 % SS of 70–133 cm$^{-3}$ in April–June and 16–57 cm$^{-3}$ in August–September. This compares well to the CCN concentrations at 0.2 % SS of 76.5 ± 28.0 cm$^{-3}$ and 39.6 ± 15.7 cm$^{-3}$ we determined in spring and summer, respectively. Overall, the aerosol at Villum appears to be slightly less hygroscopic compared to that of the Central Arctic Ocean and the Canadian Arctic. However, the measured CCN concentrations at Villum are higher than those measured in the Arctic Ocean by Martin et al. (2011) but lower than those found over the
Canadian Arctic (Lathem et al., 2013) and at Zeppelin Mountain (Jung et al., 2018). This highlights that within the Arctic region, aerosol CCN activity, and therewith probably the aerosol sources, are somewhat heterogeneous in both time and space. Additionally, the aircraft measurements, as well as Zeppelin mountain measurements, originate in the free troposphere while Villum measurements represent boundary layer sampling making the local origin of observed aerosol more likely.

During spring air mass trajectory analysis revealed lower $\kappa_{CCN}$ values for air masses from the central Arctic Ocean, compared to higher $\kappa_{CCN}$ values when air masses originated from marine and coastal areas on the Greenlandic West Coast. The latter could eventually originate from sources associated with areas of partially open sea-ice. Our trajectory analysis did not reveal different air mass origins for observed aerosols with high and low CCN activity in the summer period. This was likely to be caused by the more numerous and diverse active aerosol sources during the summer measurement period,
which again illustrates the above-described heterogeneity of aerosol hygroscopicity during the Arctic summer.

Dall'Osto et al. (2017) demonstrated the influence of partially open sea-ice on NPF frequency, with a quantitative analysis combining air mass history and sea-ice coverage. A similar approach, including the time spent over different types of Arctic surface areas combined with meteorological parameters and potential biological activity, could potentially determine individual sources of CCN-relevant aerosols in case a substantially long dataset of CCN measurements is
available.



The actual supersaturation that develops during the atmospheric updraft of an air parcel is critically important for determining which fraction of the ambient aerosol acts as CCN. Yet little is known about this process in the Arctic atmosphere. Peak supersaturations ($SS_{peak}$) were calculated by Earle et al. (2011), based on aircraft measurements of cloud properties and characterization of aerosols below cloud, over the Beaufort Sea in April. For a bimodal aerosol, which is

also well representative of our data, and under unpolluted conditions, they determined $SS_{peak}$ of 0.33–0.44 % for updraft velocities of 30–40 cm s$^{-1}$ and reported calculated κ-values of 0.2-0.3. Lower κ-values delay the onset of liquid water condensation during updraft, resulting in the aerosol population being exposed to higher $SS_{peak}$, which partly offsets the importance of κ-values. Oppositely, polluted aerosols, characterized by higher number concentrations, deplete available water vapor faster during condensation, which lowers the effective $SS_{peak}$. Values of 0.12–0.18 % SS are presented in the

above-mentioned study for polluted cases. At the subarctic station Pallas in northern Finland $SS_{peak}$ of 0.27-0.63 was determined in Sep-Oct (Anttila et al., 2012), though they observed generally lower κ-values compared to our studies at Villum, which could be explained by their proximity to anthropogenic emission sources.

Based on these peak supersaturations, the ambient $Dp_{crit}$ at Villum would be approximately 68–80 nm, yielding actual ambient CCN concentrations of about 85–92 cm$^{-3}$ and 48–53 cm$^{-3}$, during the spring and summer period, respectively.

These critical activation diameters are between the Aitken and accumulation mode (Fig. 2), indicating that even if newly formed particles grow into the lower Aitken mode, which is an established phenomenon in the Arctic atmosphere (Nguyen et al., 2016; Willis et al., 2016; Burkart et al., 2017; Dall'Osto et al., 2018a, 2018b), significant condensational growth will be required for these particles to reach cloud relevant sizes. Consequently, sources of accumulation mode aerosols could be more relevant for the CCN population compared to sources of new particle formation. This underlines the

relevance of biogenic primary marine sources because these could emit CCN-relevant aerosols directly during summer when Arctic haze accumulation mode aerosol is absent. Marine gels and water-insoluble organic material from the sea surface microlayer have been shown to be directly emitted as particles in the CCN relevant size ranges (Facchini et al., 2008; Orellana et al., 2011; Karl et al., 2013). Hamacher-Barth et al. (2016) found gel particles at Dp of predominantly >45 nm within a morphological analysis. A chemical analysis of gel polysaccharide monomers showed that pentoses and

hexoses, associated with cellular material of phytoplankton, and deoxysugars from microbial exudates were present in the Aitken and accumulation mode ranges of Arctic marine aerosols (Leck et al., 2013). A similar targeted chemical and morphological analysis of aerosol filter samples from Villum could therefore clarify the role of marine gels for late spring and summertime CCN and is proposed here for further studies.

The average RH during the spring period at Villum was 80.3 ± 7.9 %, while it was 77.3 ± 11.6 % during the summer

period. For our HTDMA measurements, we found average HGFs of 1.56 and 1.62 at 85 % RH for 120 and 240 nm particles in spring, respectively, and 1.50 at 85 % RH for 120 nm particles during summer (no valid measurements of HGFs for 240 nm were obtained in summer). The ambient RH was similar to the RH measured at Ny-Ålesund, Svalbard, reported by Rastak et al. (2014). They used calculated HGFs for 200 nm particles at 85% RH to determine the direct radiative forcing exerted by hygroscopic aerosols. For the months of April and May, they used HGFs with an average of

about 1.56, and for August, they used HGFs of about 1.59. They determined an average annual aerosol scattering enhancement factor of 4.30 ± 2.26 resulting in an aerosol-radiation RF of -0.83 to -2.60 W m$^{-2}$. Similarly, we expect particle hygroscopic growth at Villum at the ambient RH to yield a negative impact on radiative transfer as well. As we observed both higher RH and HGFs in spring compared to summer, the direct scattering effect should be more pronounced in spring and early summer compared to periods later in the year. Hygroscopic growth appears to be more pronounced in

spring, whereas CCN processes seem to be relatively more important during summer. In spring, aerosols of anthropogenic origin, dominated by acidic sulfate particles and carbonaceous aerosols, exhibit high hygroscopic growth and aerosol-radiation effects potentially are the dominating RF mechanisms during this time of year. In summer, accumulation mode aerosol concentrations are smaller, and aerosol hygroscopic growth is generally lower. In this period, natural CCN active marine aerosols could be the dominating aerosol climate forcing agent through aerosol-cloud effects. Detailed modeling

studies of the climatic influence of Arctic aerosols through aerosol-radiation and aerosol-cloud interactions could clarify this hypothesis.

Ongoing nephelometer measurements at Villum will provide a basis for further investigations of the impacts of aerosol hygroscopic growth on aerosol-radiation interactions in the nearer future. Furthermore, measurement series of hygroscopic properties of longer duration at Villum would be highly valuable, as a full yearly cycle of CCN and HGF

measurements would be a leap forward in the understanding of Arctic climate-relevant aerosol properties.



## 5    Conclusions

Arctic aerosols, and their radiation- and cloud interactions, are key factors in the changing Arctic climate. Still though, many details about their sources, hygroscopic properties, and cloud interactions are not well understood. In this work, we
provide results from two field studies conducted at the High Arctic site, Villum Research Station. One field study was carried out in the spring to early summer of 2016, while the other study was conducted from summer to autumn of the same year. During these field studies, particle hygroscopic growth factors and CCN properties were measured. Based on other studies, we expect peak supersaturations of about 0.3-0.4% to occur in Arctic clouds. This yields actual ambient average CCN concentrations of 85-92 cm$^{-3}$ and 48-53 cm$^{-3}$ during the spring and summer measurement period,
respectively. The corresponding $Dp_{crit}$ at 0.3-0.4% SS was approximately 68-80 nm, showing that accumulation mode aerosols are more important for the CCN population compared to nucleation- and Aitken mode aerosols. These results can be used directly in model parameterisation. During spring, accumulation mode aerosols originated mostly from long-range transport and Arctic haze. During summer, the concentration of accumulation mode aerosols was generally low, and we cannot present a clear indication of its origin. Sea salt influences seem to be low, and only weak indications of
primary marine organic particles were observed in the combined analysis of HGF and CCN measurements.

We did not find indications of marine influence from open pack ice based on air mass back trajectory analyses. The ambient aerosol appears to be influenced by a multitude of sources during summer. Based on other studies of the impact of hygroscopic properties on radiative forcing in the High Arctic, the hygroscopic properties at Villum very likely result in direct aerosol-radiation interactions that lead to atmospheric cooling, especially in the spring period. Future studies,
combining several additional methods, could further clarify the role of marine gels from the sea surface microlayer on CCN concentration and aerosol hygroscopic growth on the radiation balance in the High Arctic atmosphere. Targeted chemical and morphological analysis for marine gels on aerosol filter samples, aerosol scattering, or on-line aerosol composition measurements of the submicrometer aerosol could be relevant tools to further clarify open questions about the climate-relevant role of Arctic aerosols.


## 6    Acknowledgements

RL and UG were funded by the Independent Research Fund Denmark, grant DFF–4005-00482B–FTP "NUMEN". The overall study was financially supported by the Danish Environmental Protection Agency with means from the MIKA/DANCEA funds for Environmental Support to the Arctic Region, which is part of the Danish contribution to
"Arctic Monitoring and Assessment Program" (AMAP) and the Danish research project "Short lived Climate Forcers" (SLCF). We greatly acknowledge the Danish Center for the Environment for funding a grant to write this article. In addition, we want to thank our technician Bjarne Jensen and IT-expert Keld Mortensen for their great support within this study. The military staff at Station Nord is highly acknowledged for their excellent logistical help as well as Villum Foundation for funding the station infrastructure and corresponding applied instrumentation.


## 7 Data Availability

All data used in this manuscript can be found at https://doi.org/10.5281/zenodo.6787654. Sea-ice concentrations were taken from the National Sea & Ice Data Center (https://nsidc.org/data/seaice_index/archives/).




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
