# Peer review of "Measurement report: High Arctic aerosol hygroscopicity at suband supersaturated conditions during spring and summer"

_Atmospheric Chemistry and Physics, 2022_

## Author Comment (AC1)

Answer to Anonymous Referee 1:

Results of aerosol hygroscopicity measurements from a CCNC and an HTDMA are presented for spring and summer periods in 2016 at the Villum monitoring station in Greenland. These types of measurements are rare in the Arctic so the paper is a significant contribution to what is known about Arctic aerosol hygroscopicity. The research methods and analysis are thorough and the paper is well written. I only have the few comments listed below.

*We are grateful that the referee finds our data and analysis on Arctic aerosols valuable and would like to thank the referee for his/her suggestions for improvements. In the following we like to answer comments in detail. We have replied to the reviewer's comments in italics and additions/changes to the manuscript are marked in bold.*

Figure 3. Were there no UVA measurements available between August 25 and September 29 or was there no measurable UVA?

*We found that data in the named period were not available. This is why data are not shown.*

Section 3.2 and Figure 4: The sampling periods are not the same in the text and the figure. For example, the text says 20 April – 10 May and the figure says 20 April – 8 May.

*Good point, we mean May 8, this has now been changed in the text accordingly. Also, there was a small discrepancy for the summer period in the text (15.08. – 28.08. compared to 15.08. – 10.09.). This has now been corrected and changed accordingly.*

Line 441: Providing a theoretical value for ammonium bisulfate would be helpful.

*After some literature research, we unfortunately have not found a theoretical value for ammonium bisulfate. We are grateful if the reviewer can provide us with some value. As ammonium bisulfate is little more hygroscopic as ammonium sulfate, we assume that a theoretical value will be little higher. We have added this assumption in the final version of the manuscript.*

**For ammonium bisulfate we would expect a slightly higher theoretical value compared to ammonium sulfate, which would match our retrieved calculations at RH = 85 %, while our calculations at RH = 90 % show little lower values.**

Lines 552 – 566: This discussion is a little confusing. It is likely true that "the organic mass fraction must be assumed to contribute significantly to the hygroscopicity of the observed aerosol at Villum". However, the measurement techniques used focus on the particle size range where mass is negligible and number concentrations are highest. A more in-depth description of this discrepancy would be helpful along with more details on what is known about the size dependence of the organics in the Arctic. Nielsen et al. measured organics in PM1. Can more information be provided about how that organic mass is distributed across the PM1 size range? Perhaps from the Croft et al. results?

*Experimental data stating the organic mass distribution across the submicrometer size range have not been found to the best authors knowledge for a comprehensive Arctic dataset. Nielsen et al. is investigating the chemical composition at Villum with respect to the bulk PM1 size fraction. It is true that some information is found in Croft et al. for Arctic aerosols as model*

*simulations were performed supporting our arguments that the organic mass fraction can be quite high for submicrometer and even for ultrafine particles during summertime. Also, atmospheric aging does very much support that organics originated from biogenic sources can well contribute to the organic submicrometer and even ultrafine size fraction of Arctic aerosols. Also, we found a paper by Tremblay et al. (2019), who investigated the organic mass fraction compared to sulfate of submicrometer aerosol at Eureka station during particle formation events. Accordingly, we have added some text to the discussion.*

**Croft et al. (2019) performed model calculations for Arctic aerosols during summertime showing that about 50% of the mass fraction around 100 nm in diameter can be of organic origin. Also, the atmospheric aging of Arctic aerosols, which are during summer to a large extent originated from local or regional sources supports that biogenic precursors can largely contribute to the organic mass fraction of Arctic aerosols that were investigated by HTDMA and CCN counter techniques in this study. Tremblay et al. (2019) found large organic mass fractions compared to sulfate during particle formation events in the submicrometer and even ultrafine size range at Eureka station using AMS techniques.**

Line 589: "likely caused by the more numerous and diverse active aerosol sources during the summer measurement period". Perhaps this should be local aerosol sources? Spring aerosol sources can be quite diverse given the long range transport that occurs during that time of year.

*We agree that these are of course local sources compared to the spring period. Nevertheless, we argue that these sources may vary over the course of the full summer. As such we expect some heterogeneity in our observations. We have changed/added the following comment to the discussion of this topic.*

**In the summer we observed the above-described heterogeneity of aerosol hygroscopicity. This was likely to be caused by the numerous and diverse active aerosol sources during the course of the summer measurement period. These sources are mostly of local and regional origin that can stem from open waters, sea ice edges or directly from snow- and ice-covered surfaces.**

---

## Author Comment (AC2)

Answer to Anonymous Referee 2:

Massling et al. present in their manuscript observational results of aerosol hygroscopicity and aerosol cloud activation from two field campaigns carried out in spring and summer 2016 at Villum Research Station in Northern Greenland. These are valuable observations from a part of the world where observations are generally scarce but are needed to better understand aerosol-cloud and aerosol-radiation effects on Arctic climate. As such, it is important that this data and findings will be published and be accessible to the scientific community.

The paper is well written, although sometimes very lengthy and the authors are encouraged to shorten and constrain their manuscript where possible. A few but important technical details are missing and need to be added to the revised version. In addition, further clarifications (described in detail below) should be made before the manuscript can be accepted for final publication.

*We acknowledge the positive evaluation of our manuscript from the reviewer. We agree that the manuscript is partially a bit lengthy, but we have been discussing this and feel that we do not include unnecessary details, rather more the length of the manuscript results from the complexity of the dataset and the different experimental and analysis methods applied. The dataset from this remote location is extremely unique and we want to show all results and decided not to cut down some of the findings, which would be needed when shortening the paper. Please see below our detailed responses to your comments.*

Detailed comments (in chronological order):

- Abstract: The last three sentences are very general statements and partially more like an outlook. Suggest to delete them.

*We agree that these sentences are not needed especially in an abstract where predominantly facts shall be presented. We, therefore, followed the suggestion and deleted these three sentences from the abstract.*

- Introduction (2ⁿᵈ paragraph): The authors might not be aware of it, but the effect of water-uptake on particle light scattering has been actually directly measured/studied in the Arctic by Zieger et al. (2010). The scattering enhancement is indeed significantly larger compared to other continental or maritime sites (see e.g. Burgos et al, 2019) due to the special interplay between size and hygroscopicity in the Arctic (Zieger et al., 2010).

*We agree that this study should be mentioned here as it gives additional motivation for our study and points towards the importance of measuring the hygroscopicity of Arctic aerosols. We have added the following statement in the revised version of the manuscript:*

**Burgos et al. (2019) found by direct measurements of the aerosol enhancement factor that the effect of water uptake on the light scattering is higher for Arctic aerosols compared to other atmospheric reservoirs, which is based on the special interplay between size and hygroscopicity in the Arctic (Zieger et al., 2010).**

- Introduction (3rd paragraph): The regional characteristics of the Arctic (and the corresponding aerosol properties) are actually quite diverse (see e.g. discussion in Schmale at e., 2021). Certain parts of the Arctic such as the Siberian Arctic are exposed to high levels of anthropogenic/industrial activities, while the high/central Arctic e.g. shows often different seasonality in aerosols properties than the lower Arctic. This is also important for the different drivers of new particle formation (which will be addressed later in the manuscript) as e.g. discussed in detail by Schmale and Baccarini (2021). I therefore recommend that the authors should more carefully define what they actually mean with "the Arctic" for their study and also mention the regional diversity.

*We agree with this suggestion. "Arctic" is a large area and can definitively be very remote as well as close to anthropogenic sources. To more clearly define our study area and explain the possible variability, we have added the following section:*

**It should be mentioned here, that these studies concentrated on remote locations in the high Arctic. Properties of Arctic aerosols can be quite diverse depending on the exact location and time of the year (Schmale et al., 2021; Schmale & Baccarini, 2021; Schmale et al., 2022). Certain parts of the Arctic, e.g., Siberian and North American Arctic, are exposed to high levels of anthropogenic/industrial activities, while the high/central Arctic often shows different seasonality of aerosol properties compared to the lower Arctic. The exact location and seasonality also have consequences for the precursor gases of new particle formation (Schmale and Baccarini, 2021) and their subsequent growth to CCN-relevant sizes.**

- Introduction (4th paragraph): The work by Jung et al. (2018), which is cited later in the manuscript, should be mentioned here as well, since it represents a long-term study of CCNC measurements in the Arctic with the same instrumentation used here.

*We fully agree and have now included this reference also in the introduction section using the following statement:*

**Jung et al. (2018), evaluated a multi-year CCN dataset from Zeppelin Mountain, Svalbard, and found CCN concentrations at 0.2 % SS of 70–133 $cm^{-3}$ for the late spring/early summer and 16–57 $cm^{-3}$ for the late summer/early autumn period.**

- One important missing part are the details on the particle sampling. Please add a of the actual set-up sketch to the revised version. Please also add information of the used tubing (type, inner diameter, length, etc.), the inlet type (with or w/o size cut? height above ground, manufacturer, etc.).

*We agree that some important information is missing, however we feel a sketch isn't necessary, and have added the following section in the method section:*

**The SMPS sampled aerosols through a total air inlet designed and manufactured by TROPOS (Leipzig, Germany). The inlet consisted of stainless steel which sampled aerosols about 7 m above ground level and was heated to +5 °C to smoothly condition the aerosol to room air temperature where the aerosol instrumentation was placed. The total air inlet had an inner diameter of about 75 mm and was connected to an isokinetic flow splitter where the instruments were connected with ¼ inch conductive tubing. The SMPS sampled from this inlet for both campaigns while the HTDMA and CCN counter**

**sampled from a stainless-steel inlet that was designed for gases but fulfilled the requirements for particle sampling during the spring field study and for the summer field study sampled from a second total air inlet as describe above.**

- Where were the meteorological parameters measured (e.g. is the temperature shown in Fig 2 and 3 measured directly at the inlet)?

*We have added the following information:*

**Meteorological parameters were measured at a 10 m mast next to the measurement hut and originated from an average of two sensors in each height (9 m height and 3 m height).**

- Concerning the SMPS: Was a pre-impactor used? Was the SMPS data corrected for losses? If yes, how and which assumption about the particle density was used? Were the size distributions validated to a total CPC?

*Also here, we have added the following information:*

**The SMPS was operated without a pre-impactor. Losses for the ¼-inch tubing downstream of the flow splitter and in the SMPS were accounted for in the inversion routine from TROPOS (Leipzig, Germany). The SMPS has been running in parallel to a total CPC continuously for the spring field study where the measurements originating from the SMPS (integration) and total CPC agreed reasonably well (mean ratio of CPC to SMPS was 0.85). CPC measurements were unavailable for the summer field study.**

- Concerning the HTDMA: It is important to also state information of the RH of the dry DMA / selected diameter (please also add this data column to the data).

*We uploaded the data on RH of the dry DMA to Zenodo. The RH of the dry DMA was always below 15% during the spring campaign and 90% of the time below 25% during the summer campaign.*

- Line 208: As already mentioned above: Have you compared or used the total CPC to better judge which instrument was mal-functioning?

*Please see the comment above that states that the SMPS was used as the reliable source for total particle number which is why we have corrected the CCN data accordingly.*

- Line 232: Is the RH accuracy given in absolute or relative terms?

*This RH accuracy is given in absolute terms. This is a good result as RH sensors themselves are typically not better than 1% in absolute numbers. It is now clearly stated in the manuscript as the sentence has been changed accordingly.*

**We found that the set RH was reached within <2 % RH (absolute value) accuracy (Supplementary material 4).**

- Section 3.2: For a better interpretation of the size distribution data, the authors could consider to present the particle size distributions as a contour plot (e.g. hourly and normalized; it could maybe be integrated within Figure 5 and 6). This would be more consistent with the other time series and could maybe help to facilitate the interpretation of the individual features seen in Fig. 5 and 6.

*The paper does generally not focus on the size distribution and this PNSD plot is just additional information that we present. It will be very difficult to see correlations and links between the complex CCN plot and another complex three-dimensional PNSD plot. That is why the authors think it is best to present the figure on PNSD as it is.*

- I am still surprised about the extremely low uncertainties that were retrieved for the CCN concentrations in Table 1 and 2 from the curve fitting. Are they really meaningful?

*Yes they are. One must keep in mind that these values are not the standard deviation of the averages, it is the uncertainty of measurment for the CCN counter, which is low due to the long acquisition time of each sample point (5 min.).*

- Trajectory analysis: The authors mention that they have performed a trajectory analysis (e.g. pager 15, 3rd paragraph and in Sect. 4), but the results are not shown. It would indeed strengthen some on the statements and claims made later on, if this analysis would be added to the revised manuscript. Even simply calculating the time over ice, ocean and land would maybe give some more insights to the respective aerosol sources.

*We have been carefully considering this. After some discussions, we have decided to remove this part from the analysis. The reason is that no clear conclusions could be drawn from this analysis as most likely the application of air mass trajectories is more uncertain over short periods in Arctic regions because of uncertain meteorological data in remote areas. This is combined in our case with two time-limited field studies weakening the statistics. We have been working a lot with air mass back trajectories in other papers on Arctic aerosols. Here, our studies are more of a statistical nature where long-term measurements were carried out and combined with air mass back trajectories obtained for fairly long periods (i.e., several years).*

- What is the reasoning of the HTDMA to measure the growth factors at two RH close-by at 85% and 90%? Would it maybe be easier for the interpretation of the results to convert (with kappa-Köhler) all GF-values to 90%? It is not 100% clear to me on what is gained by showing both timeseries of GF at 85% and 90% in Figures 7 and 8.

*The reasoning behind the two measurement cycles is that the measurements were performed under extreme conditions in the laboratory at the Villum Research Station. HGF measurements are getting increasingly challenging when raising the RH from which they were carried out during this study. These measurements need fairly well-suited laboratories and a well-isolated instrument. For this reason, we decided to go safe by also doing measurements at 85% RH. In the end, we were managing to get the laboratory as well as the instrument stabilized temperature-wise. Converting the 85% values to 90% is easy, but would no longer show the original data, thus introducing extra uncertainty. Using our approach, we also see the differences that are produced in kappa when doing measurements at differing RH stating also the uncertainty of the overall theory as kappa is still a function of $RH_{measured}$ (HTDMA) or $SS_{measured}$ (CCN counter). In addition, one of the goals in this study was to provide values* that can be used in models and therefore reporting the HGF at two different RHs is reasonable.

- Lines 506-509: There are some recent findings that Aitken-mode particles could also be of primary origin (e.g. Xu et al., 2022 or Lawler et al., 2021), it might not be only secondary particle formation.

*This is true. We have added a paragraph here to the manuscript stating the following:*

**It has to be noted that there is also evidence for Aitken mode particles of primary origin. Several hypotheses exist which propose different production mechanisms for these particles. This could be, for example, the breakup of larger particles or the collapse of marine gel particles in droplets to a nanoparticle state (Lawler et al., 2021). It was also found that sea spray aerosols can contribute to the Aitken mode population down to sizes of 35 nm (Xu et al., 2022). Contributions from such sources to the Aitken mode aerosol at Villum cannot be fully excluded at least during the summertime, when open waters are closer to the station compared to the wintertime.**

- Page 23, 3$^{rd}$ paragraph: The authors could also reference and mention the work by Mauritsen et al. (2011) about the tenuous cloud regime in the Arctic and the susceptibility of Arctic clouds to changes in CCN concentration.

*This is a very nice paper. We cited this paper already, but in another context. We have now added to the paragraph:*

**It must be noted that the Arctic cloud regime is very susceptible to small changes in CCN concentrations (Mauritsen et al., 2011). During the summertime, particle number concentrations in Arctic environments can undergo values below 10 cm$^{-3}$ (Freud et al., 2017). This is why detailed measurements of subsaturated hygroscopicity and CCN ability are needed to understand the role of Arctic aerosols in aerosol-cloud-climate interactions.**

- Conclusions: As mentioned above, the results of the trajectory analysis are not really shown. Suggest to remove this part or add the results to the revised manuscript or SI.

*Please see the comment above that we for specific reasons removed this section from the manuscript.*

- Figure 5 and 6: Are the total particle concentrations measured by a CPC or derived from integrating the SMPS size distributions?

*The data originates from the SMPS integration, as stated in the figure description.*

- Data availability: It is great that the authors have already provided their data. I would recommend to also include the RH-data for the HTDMA (e.g. for the dry diameter, ambient, and measured at the inlet). It would also be good to clarify in the read-me if any of the data was corrected to STP (or not).

*See comment above about the RH of the dry DMA. The HTDMA and CCN counter data have not been corrected to STP, while the SMPS was. This has been stated in the read-me.*

- SI (page 1): Add "the" before the "CCN counter". Is the shown calibration a composite of all the four performed CCNC calibrations?

*This is done now. Yes, the calibration is a composite of all four calibrations. We have added to the SI the following sentence:*

**The calibration is a composite of all four performed CCNC calibrations before, during, and after the field studies.**

Minor comments:

- Line 23: Add "the" before "initial"

*Done.*

- Line 565: take -> taking; remove one of the "only"s

*Done.*

- Line 605: Add "%" behind 0.63

*Done.*

- Line 397: Suggest to remove the "substantial" or clarify what you mean with this.

*It is removed now.*

References (now included in the manuscript if there were not already included before):

Burgos, M., Andrews, E., Titos, G., Alados-Arboledas, L., Baltensperger, U., Day, D., Jefferson, A., Kalivitis, N., Mihalopoulos, N., Sherman, J., Sun, J., Weingartner, E., and Zieger, P.: A global view on the effect of water uptake on aerosol particle light scattering, Scientific Data, 6, 157, https://doi.org/10.1038/s41597-019-0158-7, 2019.

Jung, C. H., Yoon, Y. J., Kang, H. J., Gim, Y., Lee, B. Y., Ström, J., Krejci, R., Tunved, P.: The seasonal characteristics of cloud condensation nuclei (CCN) in the arctic lower troposphere, Tellus B: Chemical and Physical Meteorology,70, 1-13, https://doi.org/10.1080/16000889.2018.1513291, 2018.

Lawler, M. J., Saltzman, E. S., Karlsson, L., Zieger, P., Salter, M., Baccarini, A., et al. (2021). New insights into the composition and origins of ultrafine aerosol in the summertime high Arctic. Geophysical Research Letters, 48(21), 1–11. https://doi.org/10.1029/2021GL094395

Mauritsen, T., Sedlar, J., Tjernström, M., Leck, C., Martin, M., Shupe, M., et al. (2011). An Arctic CCN-limited cloud-aerosol regime. Atmos-pheric Chemistry and Physics, 11(1), 165–173. https://doi.org/10.5194/acp-11-165-2011

Schmale, J., Zieger, P., & Ekman, A. M. L. (2021). Aerosols in current and future Arctic climate. Nature Climate Change, 11(2), 95–105. https://doi.org/10.1038/s41558-020-00969-5

Schmale, J., & Baccarini, A. (2021). Progress in unraveling atmospheric new particle formation and growth across the Arctic. Geophysical Research Letters, 48, e2021GL094198. https://doi.org/10.1029/2021GL094198

Xu, W., Ovadnevaite, J., Fossum, K.N. et al. Sea spray as an obscured source for marine cloud nuclei. Nat. Geosci. 15, 282–286 (2022). https://doi.org/10.1038/s41561-022-00917-2

Zieger, P., Fierz-Schmidhauser, R., Gysel, M., Ström, J., Henne, S., Yttri, K. E., Baltensperger, U., and Weingartner, E.: Effects of relative humidity on aerosol light scattering in the Arctic, Atmos. Chem. Phys., 10, 3875–3890, https://doi.org/10.5194/acp-10-3875-2010, 2010.

---

## Editor Decision (ED1)

**Main comment**

Previous referee comment on previous version:
Line 441: Providing a theoretical value for ammonium bisulfate would be helpful.

After some literature research, we unfortunately have not found a theoretical value for ammonium bisulfate. We are grateful if the reviewer can provide us with some value. As ammonium bisulfate is little more hygroscopic as ammonium sulfate, we assume that a theoretical value will be little higher. We have added this assumption in the final version of the manuscript.

Editor comment: Even if there is indeed no kappa directly given in the literature, the authors should be able to derive it based on the underlying equations and corresponding literature values. E.g. using measured growth factors and their equation 4 or using the estimate of the water activity, e.g. (Tang and Munkelwitz, 1994), and the equation set by (Petters and Kreidenweis, 2007) should allow an estimate of the kappa for ammonium bisulfate.
The sentence in l. 453 could be replaced accordingly.

**Technical comments**

l. 18: Here and at other places in the manuscript, it seems redundant to write 'kappa' and $\kappa$. Please either write out 'kappa' OR use the Greek Letter $\kappa$, not both.

l. 40: why 'or ability to uptake water'? Isn't this the definition of hygroscopicity and thus should read then, 'i.e. the ability to take up water'?

Figures 2 and 3: Correct the spelling of 'precipitation' on the y-axis and legend of panels a.

l. 356: Please add a reference for 'EBAS data base'.

l. 530: 'CCN and HTDMA hygroscopity' sounds rather colloquial. You describe 'particle hygroscopicity derived based on CCN and HTDMA measurements' . Please replace the header accordingly by this or similar wording.

---

## Author Response (AR2)

*Dear editor,*

*we are grateful for your valuable comments.*

Main comment

Previous referee comment on previous version:
Line 441: Providing a theoretical value for ammonium bisulfate would be helpful.

After some literature research, we unfortunately have not found a theoretical value for ammonium bisulfate. We are grateful if the reviewer can provide us with some value. As ammonium bisulfate is little more hygroscopic as ammonium sulfate, we assume that a theoretical value will be little higher. We have added this assumption in the final version of the manuscript.

Editor comment:
Even if there is indeed no kappa directly given in the literature, the authors should be able to derive it based on the underlying equations and corresponding literature values. E.g. using measured growth factors and their equation 4 or using the estimate of the water activity, e.g. (Tang and Munkelwitz, 1994), and the equation set by (Petters and Kreidenweis, 2007) should allow an estimate of the kappa for ammonium bisulfate. The sentence in l. 453 could be replaced accordingly.

*We have taken your main comment into account and calculated the $\kappa$-values using the Extended AIM Aerosol Thermodynamics Model. The text now reads as follows:*

**In general, relatively high HGFs were found for the spring campaign assuming ammonium sulfate or ammonium bisulfate being a major compound of the observed aerosol. While $\kappa_{HTDMA}$ mean values range between 0.54 and 0.60 at RH = 85 % and between 0.46 and 0.51 at RH = 90 % (Table 4), for comparison a theoretical value for ammonium sulfate is calculated as 0.49 at RH = 89.7 % and 0.53 at RH 84.8 % using the Extended AIM Aerosol Thermodynamics Model (Clegg et al., 1998). Calculated values using the same model for ammonium bisulfate are 0.56 at RH = 89.8 % and 0.61 at RH = 84.9 %.**

Technical comments

*We have also addressed all the technical suggestions you made.*

l. 18: Here and at other places in the manuscript, it seems redundant to write 'kappa' and ⬚. Please either write out 'kappa' OR use the Greek Letter ⬚, not both.

*This is changed and the Greek Letter is used throughout the whole manuscript.*

l. 40: why 'or ability to uptake water'? Isn't this the definition of hygroscopicity and thus should read then, 'i.e. the ability to take up water'?

*This is changed now. The new text reads as follows:*

**"… which is the ability to take up water.**"

Figures 2 and 3: Correct the spelling of 'precipitation' on the y-axis and legend of panels a.

*This is changed now.*

l. 356: Please add a reference for 'EBAS data base'.

*A reference is added now:*

**(https://ebas.nilu.no/)**

l. 530: 'CCN and HTDMA hygroscopity' sounds rather colloquial. You describe 'particle hygroscopicity derived based on CCN and HTDMA measurements' . Please replace the header accordingly by this or similar wording.

*The header has been replaced now. The new header reads as follows:*

**Comparison of particle hygroscopicity derived based on CCN and HTDMA measurements**